# VQ-Transplant: Efficient VQ-Module Integration for Pre-trained Visual Tokenizers

**Xianghong Fang**[1*]   **Yuan Yuan**[2]   **Dehan Kong**[1*]   **Tim G. J. Rudner**[1,3*]
[1]University of Toronto    [2]Boston College    [3]Vijil

## ABSTRACT

Vector Quantization (VQ) underpins modern discrete visual tokenization. However, training quantization modules for state-of-the-art VQ-based models requires significant computational resources which, in practice, all but prevents the development of novel, cutting-edge VQ techniques under resource constraints. To address this limitation, we propose **VQ-Transplant**, a simple framework that enables plug-and-play integration of new VQ modules into frozen, pre-trained tokenizers by replacing their native VQ modules. Crucially, the proposed transplantation process preserves all encoder-decoder parameters, obviating the need for costly end-to-end retraining when modifying the quantization method. To mitigate decoder-quantization mismatch, we introduce a lightweight decoder adaptation strategy (trained for only 5 epochs on ImageNet-1k) to align feature priors with the new quantization space. In our empirical evaluation, we find that VQ-Transplant allows obtaining near state-of-the-art reconstruction fidelity for industry-level models like VAR while reducing the training cost by 95%. VQ-Transplant democratizes quantization research by enabling resource-efficient integration of novel VQ techniques while matching industry-level reconstruction performance. Code and models are available at `VQ-Transplant`.

## 1 INTRODUCTION

Vector Quantization (VQ) is a cornerstone of modern discrete visual tokenization frameworks, enabling efficient learning of discrete representations critical for downstream tasks including visual generation (van den Oord et al., 2017; Esser et al., 2021; Lee et al., 2022; Sun et al., 2024; Tian et al., 2024) and vision-language modeling (Ramesh et al., 2021; Bao et al., 2022; Li et al., 2024; Wu et al., 2024). State-of-the-art visual tokenizers such as VQGAN (Esser et al., 2021; Sun et al., 2024) and VAR (Sun et al., 2024) leverage adversarial training (Goodfellow et al., 2014) to achieve exceptional reconstruction fidelity by aligning synthesized outputs with real-data distributions.

However, achieving state-of-the-art fidelity through adversarial training requires substantial computational resources to enable training on large-scale datasets like ImageNet (Deng et al., 2009) and OpenImages (Kuznetsova et al., 2018), as detailed in Table 1. Moreover, adversarial training is inherently unstable (Isola et al., 2017; Karras et al., 2019), further complicating the process. These challenges severely limit the exploration of novel quantization algorithms, particularly in resource-constrained environments where end-to-end training of full encoder-decoder architectures is computationally infeasible.

This practical constraint raises a critical yet underexplored question:

> *Can we decouple the development of VQ methods from the computational*
> *overhead associated with training tokenizers from scratch?*

Current approaches (Esser et al., 2021; Sun et al., 2024; Tian et al., 2024; Li et al., 2025; Han et al., 2025) treat the VQ module and the encoder-decoder framework as a monolithic system, necessitating joint optimization of all components. Such tight integration not only significantly increases computational cost but also impedes the integration of novel quantization techniques into pre-trained models. For instance, replacing a frozen tokenizer's native VQ module with an improved

---

Corresponding authors: `xianghong.fang@mail.utoronto.ca`, `dehan.kong@utoronto.ca`, and `tim.rudner@utoronto.ca`.

quantization variant may lead to decoder-quantization distributional mismatch since the decoder's conditioning depends on the original quantization space.

In this paper, we propose **VQ-Transplant**, a computationally efficient framework enabling plug-and-play replacement of arbitrary VQ algorithms within pre-trained visual tokenizers (e.g., VAR (Tian et al., 2024)), without costly end-to-end retraining from scratch. Our key insight is to preserve the frozen encoder-decoder parameters of state-of-the-art tokenizers while replacing their native VQ modules with superior alternatives. To address distributional discrepancies between new quantization spaces and frozen decoders, we propose a lightweight decoder adaptation strategy—requiring only five epochs of training on ImageNet-1k—to align the decoder's learned feature priors with the new quantizer's latent space. This decoupling eliminates the need for prohibitive full-model retraining, allowing cheap and rapid iterations in the development of novel VQ algorithms.

We demonstrate the effectiveness of the VQ-Transplant framework by evaluating multiple VQ approaches for visual tokenization tasks. Building upon distributional alignment principles (Fang et al., 2025), we introduce **MMD VQ**, which leverages maximum mean discrepancy to align the distributions of feature and codebook vectors, to improve compatibility with VQ-Transplant. For example, when integrating MMD VQ into the pre-trained VAR tokenizer (Tian et al., 2024), VQ-Transplant achieves superior reconstruction fidelity (0.81 rFID) while being $21.8\times$ faster than training vanilla VAR (0.92 rFID), as shown in Table 1.

Table 1: Comparison of computational cost between different tokenizers.

| Tokenizers | Dataset | GPUs | Training Hours | Speedup |
|---|---|---|---|---|
| Llama GEN (Sun et al., 2024) | ImageNet-1k | $2 \times$ A100 | 200 | 9.1 |
| ImageFolder (Li et al., 2025) | ImageNet-1k | $32 \times$ A100 | 40 | 29.1 |
| VAR (Tian et al., 2024) | OpenImages | $16 \times$ A100 | 60 | 21.8 |
| UniTok (Ma et al., 2025) | OpenImages | $256 \times$ A100 | 50 | 290.9 |
| **VQ-Transplant (Ours)** | ImageNet-1k | $2 \times$ A100 | 22 | - |

Our key contributions are as follows:

1. Our primary contribution is **VQ-Transplant**, a computationally efficient framework for discrete visual tokenization designed to enable cheap and repid exploration of novel Vector Quantization (VQ) techniques. By enabling seamless plug-and-play integration of new quantization algorithms into pre-trained tokenizers, VQ-Transplant obviates resource-intensive retraining, thereby significantly lowering the barrier to innovation in VQ research.

2. Our secondary contribution is **MMD-VQ**, a novel VQ method introduced within this framework. Specifically designed to enable improved compatibility with VQ-Transplant, MMD VQ leverages maximum mean discrepancy to achieve distributional alignment. This method demonstrates superior reconstruction fidelity compared to the vanilla VAR approach, further validating the effectiveness and versatility of the VQ-Transplant framework.

## 2 RELATED WORK

**Visual Tokenizer for Generative Models.** The rapid evolution of generative models has catalyzed substantial interest in visual tokenization, a pivotal component that bridges raw visual signals and latent representations for synthesis. Contemporary visual generative models predominantly adhere to two paradigms (Wang et al., 2024): language model-based and diffusion-based approaches. The former harnesses sequence modeling to frame visual generation as next-token prediction (van den Oord et al., 2017; Esser et al., 2021; Yu et al., 2023; Sun et al., 2024; Ma et al., 2025), relying on discrete tokenizers such as VQVAE (van den Oord et al., 2017). In contrast, diffusion models (Ho et al., 2020; Song et al., 2021a;b) utilize continuous tokenizers (e.g., VAEs (Kingma & Welling, 2014; Rombach et al., 2022; Chen et al., 2025a;b; Xie et al., 2025)) to encode images into latent distributions.

**Adversarial Training in Discrete Visual Tokenizers.** Visual tokenizers define the expressivity boundary of generative systems (Esser et al., 2021; Yan et al., 2021; Hong et al., 2023). While early VQVAEs (van den Oord et al., 2017) were plagued by over-smoothed reconstructions and suboptimal perceptual quality, modern discrete tokenizers (Esser et al., 2021; Li et al., 2025; Tian et al., 2024) leverage adversarial training techniques, such as PatchGAN (Isola et al., 2017) and StyleGAN (Karras et al., 2019), to enhance reconstruction fidelity by aligning reconstructed outputs with real data distributions. However, such adversarial training is computationally intensive, often requiring extensive training on large-scale datasets like ImageNet (Deng et al., 2009) or OpenImages (Kuznetsova et al., 2018), as detailed in Table 1, and is prone to optimization instability. These challenges hinder researchers exploring novel VQ algorithms, particularly those with limited computational resources.

To address this training burden, we introduce VQ-Transplant, a computationally efficient framework that enables plug-and-play integration of arbitrary VQ algorithms into pre-trained visual tokenizers, circumventing the need for costly end-to-end retraining from scratch.

**Vector Quantization in Discrete Visual Tokenizers.** As the core component in discrete visual tokenizer, VQ acts as a compressor that discretizes continuous latent features into discrete visual tokens by mapping them to the nearest code vectors within a learnable codebook, utilizing the corresponding code indices as visual tokens. Despite its widespread adoption, vanilla VQ (van den Oord et al., 2017) suffers from critical limitations, including training instability and codebook collapse. Recent efforts(Huh et al., 2023; Razavi et al., 2019; Zheng & Vedaldi, 2023) have sought to mitigate these issues through enhanced codebook update mechanisms. However, their effectiveness remains heavily dependent on meticulous codebook initialization, thereby limiting their robustness and generalizability (Fang et al., 2025). To address these challenges, Fang et al. (2025) introduced Wasserstein VQ, a theoretically principled approach that reformulates codebook learning as the problem of distribution matching between code and feature vectors. Nevertheless, Wasserstein VQ critically relies on Gaussian distribution assumptions. When feature vectors in real-world applications deviate from Gaussianity, Wasserstein VQ reduces to merely aligning first- and second-order statistics between features and code vectors, failing to achieve effective distribution alignment.

## 3 PRELIMINARIES: DISCRETE VISUAL TOKENIZERS

A discrete visual tokenizer typically consists of three key components: an encoder $\mathcal{E}_\theta$, a VQ module $\mathcal{Q}_\phi$, and a decoder $\mathcal{D}_\varphi$. Given an input image $\boldsymbol{x} \in \mathbb{R}^{H \times W \times 3}$, the encoder $\mathcal{E}_\theta$ first produces a set of $d$-dimensional feature embeddings $\boldsymbol{z}_e = \mathcal{E}_\theta(\boldsymbol{x}) \in \mathbb{R}^{\frac{H}{f} \times \frac{W}{f} \times d}$, with a spatial downsampling factor of $f \times f$. The VQ module then discretizes these continuous features through nearest-neighbor codebook lookup, where each spatial feature $\boldsymbol{z}_e^{ij} \in \mathbb{R}^d$ maps to its closest codebook entry $\boldsymbol{e}_k$:

$$r^{ij} = \arg\min_k \|\boldsymbol{z}_e^{ij} - \boldsymbol{e}_k\|_2^2, \tag{1}$$

yielding spatial visual token $r^{ij} \in \mathbb{Z}^+$. The quantized latent representation $\boldsymbol{z}_q^{ij} = \mathcal{Q}_\phi(\boldsymbol{z}_e^{ij}) = \boldsymbol{e}_{r^{ij}}$ is then decoded to reconstruct the image $\widehat{\boldsymbol{x}} = \mathcal{D}_\varphi(\boldsymbol{z}_q)$.

While VQVAE (van den Oord et al., 2017) demonstrated the feasibility of discrete visual tokenization, generated images in early work exhibited oversmoothed reconstructions and suboptimal perceptual fidelity. Subsequent improvements (Esser et al., 2021; Li et al., 2025; Tian et al., 2024) have integrated adversarial training (Isola et al., 2017; Karras et al., 2019) and VGG-based perceptual regularization (Simonyan & Zisserman, 2015). The VQGAN framework (Esser et al., 2021) optimizes a composite objective:

$$\mathcal{L}(\theta, \phi, \varphi) = \|\widehat{\boldsymbol{x}} - \boldsymbol{x}\|_2^2 + \beta\|\text{sg}(\boldsymbol{z}_q) - \boldsymbol{z}_e\|_2^2 + \|\text{sg}(\boldsymbol{z}_e) - \boldsymbol{z}_q\|_2^2 + \lambda_P\mathcal{L}_{\text{Per}} + \lambda_G\mathcal{L}_{\text{GAN}}, \tag{2}$$

where $\text{sg}(\cdot)$ is the stop-gradient operation, $\mathcal{L}_{\text{Per}}$ computes feature-space discrepancies using a frozen VGG network (Zhang et al., 2018), and $\mathcal{L}_{\text{GAN}}$ uses hinge-based adversarial losses (Lim & Ye, 2017). Hyperparameters $\beta$, $\lambda_P$ and $\lambda_G$ balance the losses. Recent VQGAN-based methods (Sun et al., 2024; Tian et al., 2024; Li et al., 2025) achieve state-of-the-art reconstruction fidelity. However, their reliance on adversarial training incurs substantial computational costs, as shown in the Table 1.

## 4 THE VQ-TRANSPLANT FRAMEWORK

### 4.1 TWO-STAGE VQ-TRANSPLANT

To mitigate the computational overhead of end-to-end retraining, we propose **VQ-Transplant**, a computationally efficient framework enabling plug-and-play replacement of arbitrary VQ modules within pre-trained visual tokenizers. Our method has two parts: (1) VQ module substitution and (2) decoder adaptation, as detailed below.

**Stage I: VQ Module Substitution,** Given a pretrained discrete visual tokenizer with encoder $\mathcal{E}_{\theta^*}$, decoder $\mathcal{D}_{\varphi^*}$ and native VQ module $\mathcal{Q}_{\phi^*}^{\text{pretrain}}$, we substitute $\mathcal{Q}_{\phi^*}^{\text{pretrain}}$ with a new VQ module $\mathcal{Q}_\phi^{\text{new}}$ while freezing $\theta^*$ and $\varphi^*$. Let $\boldsymbol{z}_e = \mathcal{E}_{\theta^*}(\boldsymbol{x})$ denote the encoder's latent embedding, and $\boldsymbol{z}_q(\phi) = \mathcal{Q}_\phi^{\text{new}}(\boldsymbol{z}_e)$ represent the quantized latent from the new VQ module. The training objective for $\mathcal{Q}_\phi^{\text{new}}$ is:

$$\mathcal{L}_{\text{VQ}}(\phi) = \|\text{sg}(\boldsymbol{z}_e) - \boldsymbol{z}_q(\phi)\|_2^2 + \gamma\mathcal{L}_{\text{unique}}(\mathcal{Q}_\phi^{\text{new}}), \tag{3}$$

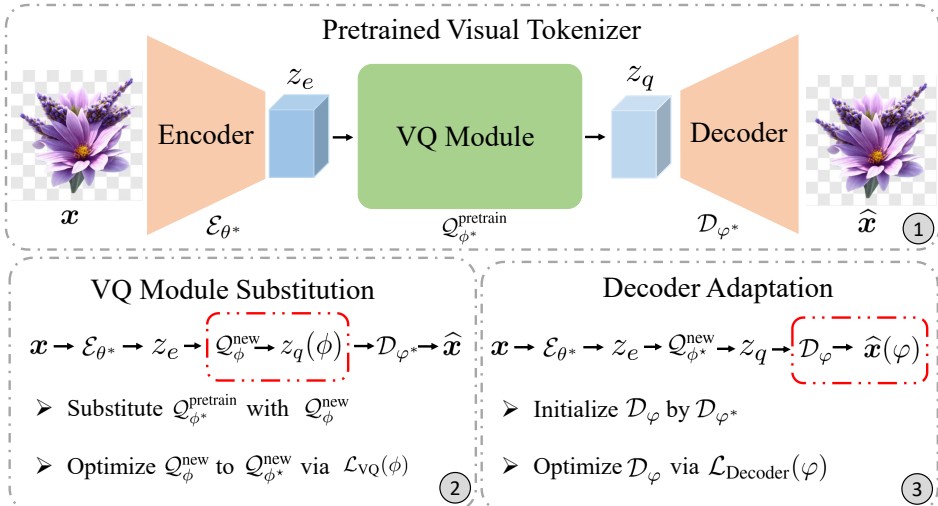

Figure 1: Illustration of the **VQ-Transplant** framework. Block 1 represents a pretrained visual tokenizer with three key components: an encoder, a decoder, and a native VQ module. Block 2 and 3 denote the VQ module substitution and decoder adaptation stages in the VQ-Transplant framework.

where $\mathcal{L}_{\text{unique}}$ is a uniqueness-enforcing loss in new VQ modules (e.g., Wasserstein loss for Wasserstein VQ (Fang et al., 2025)), and $\gamma$ balances the terms. This stage ensures minimization of the quantization error while accounting for its inherent constraints.

**Stage II: Decoder Adaptation.** While Stage I might effectively reduces quantization error, the frozen decoder $\mathcal{D}_{\varphi^*}$ remains suboptimal for reconstructing inputs from the updated quantized representations $z_q(\phi)$ due to mismatch between the decoder and the quantized space. To address this incompatibility, we propose a lightweight decoder adaptation scheme that preserves the frozen encoder $\mathcal{E}_{\theta^*}$, and optimized VQ module $\mathcal{Q}_{\phi^\star}^{\text{new}}$ while updating only the decoder parameters. Specifically, we compute quantized features as $z_q = \mathcal{Q}_{\phi^\star}^{\text{new}}(\mathcal{E}_{\theta^*}(x))$ and reconstruct the image via $\widehat{x}(\varphi) = \mathcal{D}_\varphi(z_q)$, where $\varphi$ is initialized from $\varphi^*$. The decoder is optimized through a composite loss:

$$\mathcal{L}_{\text{Decoder}}(\varphi) = \|\widehat{x}(\varphi) - x\|_2^2 + \lambda_P \mathcal{L}_{\text{Per}}(\varphi) + \lambda_G \mathcal{L}_{\text{GAN}}(\varphi), \tag{4}$$

where the hyperparameters $\lambda_P$ and $\lambda_G$ balance the terms. We follow Tian et al. (2024), Chen et al. (2025a), and Li et al. (2025) and employ an identical frozen DINO-S (Caron et al., 2021; Oquab et al., 2024) discriminator, which shares a similar architecture to StyleGAN (Karras et al., 2020; 2019). To improve discriminator training, we incorporate DiffAug (Zhao et al., 2020), consistency regularization (Zhang et al., 2019), and LeCAM regularization (Tseng et al., 2021) as implemented in Tian et al. (2024). We also discuss another optimization approach in Appendix C, where the encoder, decoder, and VQ module are jointly optimized, instead of optimizing the decoder alone.

## 4.2 MMD-VQ: Distribution-Aligned Vector Quantization

To enhance compatibility with the VQ-Transplant framework, we propose MMD-VQ, a novel vector quantization method that leverages Maximum Mean Discrepancy (MMD; Gretton et al., 2012; Sriperumbudur et al., 2009) to align the distributions of underlying features and the codebook. The motivation for using MMD is grounded in a well-established theoretical principle: a vector quantizer approaches optimality when its codebook distribution closely matches the feature distribution, minimizing quantization error while maximizing codebook utilization, as rigorously supported by prior work (Fang et al., 2025; Graf & Luschgy, 2000). MMD achieves this alignment through characteristic kernels that guarantee universal distribution matching. Unlike previous Gaussian-dependent approaches (Fang et al., 2025), MMD-VQ makes no parametric assumptions and robustly aligns feature and codebook distributions even for complex, non-Gaussian data. This flexibility makes MMD-VQ particularly suitable for the VQ-Transplant framework, especially in scenarios where feature distributions deviate from idealized parametric forms, such as multi-modal, heavy-tailed, or otherwise non-Gaussian distributions. A more detailed discussion of the motivation and advantages of MMD-VQ is provided in Appendix B.

Specifically, we collect feature vectors $X = \{z_1, z_2, ..., z_N\}$ (spatial features $\boldsymbol{z}_e^{ij}$) and codebook vectors $Y = \{e_1, e_2, ..., e_K\}$. The squared MMD distance is defined as:

$$\mathcal{D}^2_{\text{MMD}}(X, Y) = \frac{1}{N^2} \sum_{i=1}^{N} \sum_{j=1}^{N} k(z_i, z_j) + \frac{1}{K^2} \sum_{i=1}^{K} \sum_{j=1}^{K} k(e_i, e_j) - \frac{2}{NK} \sum_{i=1}^{N} \sum_{j=1}^{K} k(z_i, e_j), \quad (5)$$

where $k(\cdot, \cdot)$ denotes a characteristic kernel. Crucially, $\mathcal{D}^2_{\text{MMD}}(X, Y) = 0$ iff $\mathcal{P}_X = \mathcal{P}_Y$, making MMD a useful nonparametric divergence metric. We employ a multi-Gaussian kernel $k(x, y) = \sum_i \exp(-\|x - y\|^2 / 2\sigma_i^2)$ and define $\mathcal{L}_{\text{unique}} = \mathcal{D}^2_{\text{MMD}}(X, Y)$ in Equation (3) to achieve distribution alignment.

## 5 EMPIRICAL EVALUATION

We present a comprehensive empirical evaluation of VQ-Transplant from three perspectives. First, we investigate the integration of multi-scale VQ algorithms within VQ-Transplant in Section 5.1. Second, we evaluate fixed-scale VQ algorithms under the VQ-Transplant framework in Section 5.2. Both studies are conducted using the ImageNet-1k dataset (Deng et al., 2009). Finally in Section 5.3, we examine the generalization capability of VQ-Transplant across diverse datasets to further illustrate the strengths and potential of the proposed framework.

**Experiment Setup.** This work investigates the VQ-Transplant framework using a pre-trained VAR tokenizer (Tian et al., 2024) throughout all experiments. During VQ module substitution, we freeze the encoder-decoder parameters of the pre-trained model and replace its native multi-scale VQ module with either new multi-scale VQ modules or fixed-scale VQ modules. For the latter approach, we implement a parallel quantization system wherein 32-dimensional feature vectors are partitioned into two 16-dimensional sub-vectors. These sub-vectors undergo independent quantization via separate VQ modules before concatenation to form the final 32-dimensional vectors for decoder input. During decoder adaptation, we freeze parameters for both the encoder and transplanted VQ modules. We implemented the VQ-transplant using five distinct quantization algorithms: Vanilla VQ (van den Oord et al., 2017), EMA VQ (Razavi et al., 2019), Online VQ (Zheng & Vedaldi, 2023), Wasserstein VQ (Fang et al., 2025), and our proposed MMD VQ, adapted respectively for both multi-scale and fixed-scale configurations in the pretrained VAR tokenizer. Additional implementation details are provided in Appendix A.

**Evaluation Metrics.** To quantify VQ performance, we report the quantization error ($\mathcal{E}$) and codebook utilization ($\mathcal{U}$). For reconstruction quality, we measure: peak signal-to-noise ratio (PSNR), structural similarity index (SSIM), Fréchet Inception Distance (r-FID; Heusel et al., 2017), perceptual similarity (LPIPS; Zhang et al., 2018), and reconstruction inception score (r-IS; Salimans et al., 2016).

**Baselines.** We compare the proposed MMD VQ within VQ-Transplant framework with a set of baselines: DQVAE (Huang et al., 2023a), DF-VQGAN (Ni et al., 2022), DiVAE (Shi et al., 2022), RQVAE (Lee et al., 2022), VQGAN (Esser et al., 2021), VQGAN-FC (Yu et al., 2022), VQGAN-EMA (Razavi et al., 2019), VQGAN-LC (Zhu et al., 2024), Llama GEN (Sun et al., 2024), and VAR (Tian et al., 2024), all evaluated on the ImageNet-1k dataset.

**Main Results.** As demonstrated in Table 2, our VQ-Transplant framework equipped with MMD-VQ and MMD-VAR outperform competing baselines in critical reconstruction fidelity metrics, including r-FID and r-IS. This validates the framework's capacity to deliver industry-level reconstruction fidelity when integrated with compatible quantization algorithms like MMD VQ and MMD-VAR. Beyond superior reconstruction fidelity, our VQ-Transplant implementations achieve significant efficiency gains: As quantified in Table 1, MMD VQ and MMD VAR demonstrate $21.8\times$ faster training than standard VAR (Tian et al., 2024) while simultaneously exceeding the reconstruction performance of the original VAR tokenizer. Specifically, MMD VQ achieves 0.86 r-FID and MMD VAR reaches 0.81 r-FID, outperforming the baseline VAR tokenizer's 0.92 r-FID.

### 5.1 INTEGRATION OF MULTI-SCALE VECTOR QUANTIZATION ALGORITHMS

We investigate the integration of five multi-scale VQ algorithms within the VQ-Transplant framework: Vanilla VAR, EMA VAR, Online VAR, Wasserstein VAR, and MMD VAR. As shown in Table 3, when

Table 2: Reconstruction performance on the ImageNet-1K dataset. "FS VQ" indicates fixed-scale quantization methods, while "MS VQ" corresponds to multi-scale quantization methods. [†]: results cited from VQGAN-LC (Zhu et al., 2024); [⋆]: results cited from Llama GEN (Sun et al., 2024).

| Method | VQ Types | Tokens | Codebook Size $K$ | $\mathcal{U}$ (↑) | r-FID ↓ | r-IS ↑ | LPIPS ↓ | PSNR ↑ | SSIM ↑ |
|---|---|---|---|---|---|---|---|---|---|
| DQVAE[†] | FS VQ | 256 | 1024 | - | 4.08 | - | - | - | - |
| DiVAE[†] | FS VQ | 256 | 16384 | - | 4.07 | - | - | - | - |
| RQVAE[†] | FS VQ | 256 | 16384 | - | 3.20 | - | - | - | - |
| RQVAE[†] | FS VQ | 512 | 16384 | - | 2.69 | - | - | - | - |
| RQVAE[†] | FS VQ | 1024 | 16384 | - | 1.83 | - | - | - | - |
| DF-VQGAN[†] | FS VQ | 256 | 12288 | - | 5.16 | - | - | - | - |
| DF-VQGAN[†] | FS VQ | 1024 | 8192 | - | 1.38 | - | - | - | - |
| Llama GEN[⋆] | FS VQ | 256 | 16384 | 97.0% | 2.19 | - | - | 20.79 | 67.5 |
| VQGAN[†] | FS VQ | 256 | 16384 | 3.4% | 5.96 | - | 0.17 | 23.3 | 52.4 |
| | FS VQ | 256 | 50000 | 1.1% | 5.44 | - | 0.17 | 22.5 | 52.5 |
| | FS VQ | 256 | 100000 | 0.5% | 5.44 | - | 0.17 | 22.3 | 52.5 |
| VQGAN-FC[†] | FS VQ | 256 | 16384 | 11.2% | 4.29 | - | 0.17 | 22.8 | 54.5 |
| | FS VQ | 256 | 50000 | 3.6% | 4.96 | - | 0.15 | 23.1 | 54.7 |
| | FS VQ | 256 | 100000 | 1.9% | 4.65 | - | 0.15 | 22.9 | 55.1 |
| VQGAN-EMA[†] | FS VQ | 256 | 16384 | 83.2% | 3.41 | - | 0.14 | 23.5 | 56.6 |
| | FS VQ | 256 | 50000 | 40.2% | 3.88 | - | 0.14 | 23.2 | 55.9 |
| | FS VQ | 256 | 100000 | 24.2% | 3.46 | - | 0.13 | 23.4 | 56.2 |
| VQGAN-LC[†] | FS VQ | 256 | 16384 | **99.9%** | 3.01 | - | 0.13 | 23.2 | 56.4 |
| | FS VQ | 256 | 50000 | **99.9%** | 2.75 | - | 0.13 | 23.8 | 58.4 |
| | FS VQ | 256 | 100000 | **99.9%** | 2.62 | - | 0.12 | 23.8 | 58.9 |
| **MMD VQ (Ours)** | FS VQ | 512 | 16384 | 99.8% | 1.05 | 191.2 | 0.115 | 24.31 | 63.7 |
| | FS VQ | 512 | 32768 | **99.9%** | 0.97 | 194.1 | 0.110 | 24.53 | 64.7 |
| | FS VQ | 512 | 65536 | **99.9%** | **0.86** | **197.1** | 0.106 | 24.65 | 65.0 |
| VAR | MS VQ | 680 | 4096 | **100%** | 0.92 | 198.6 | **0.100** | **24.37** | **63.9** |
| **MMD VAR (Ours)** | MS VQ | 680 | 4096 | **100%** | 0.91 | 199.2 | 0.108 | 24.16 | 63.2 |
| **MMD VAR (Ours)** | MS VQ | 680 | 8192 | **100%** | **0.81** | **201.0** | 0.104 | **24.37** | 63.8 |

Table 3: Reconstruction performance of multi-scale VQ algorithms on ImageNet-1K dataset. For each codebook size and each phase, the best-performing result is highlighted in bold.

| Methods | Phase | Tokens | Codebook Size $K$ | $\mathcal{E}$(↓) | $\mathcal{U}$ (↑) | PSNR(↑) | SSIM(↑) | LPIPS (↓) | r-FID(↓) | r-IS(↑) |
|---|---|---|---|---|---|---|---|---|---|---|
| VAR Tokenizer w/o VQ | - | - | - | 0 | - | 23.31 | 67.7 | 0.140 | 9.71 | 134.9 |
| VAR Tokenizer (Tian et al., 2024) | - | 680 | 4096 | 0.283 | 100% | 24.37 | 63.9 | **0.100** | 0.92 | 198.6 |
| Vanilla VAR | Substitution | 680 | 4096 | 0.305 | 38.2% | 23.53 | 61.2 | 0.137 | 1.84 | 180.3 |
| EMA VAR | Substitution | 680 | 4096 | 0.321 | 99.9% | 23.34 | 60.0 | 0.151 | 2.28 | 171.8 |
| Online VAR | Substitution | 680 | 4096 | 0.276 | 99.0% | 24.16 | 63.2 | 0.124 | 1.56 | 186.9 |
| Wasserstein VAR | Substitution | 680 | 4096 | **0.255** | **100%** | 24.49 | 64.3 | 0.117 | 1.57 | 188.6 |
| MMD VAR | Substitution | 680 | 4096 | **0.255** | **100%** | **24.52** | **64.4** | **0.116** | **1.52** | **189.4** |
| Vanilla VAR | Substitution | 680 | 8192 | 0.309 | 22.9% | 23.29 | 61.0 | 0.139 | 1.93 | 179.3 |
| EMA VAR | Substitution | 680 | 8192 | 0.312 | 99.8% | 23.45 | 60.5 | 0.147 | 2.18 | 173.6 |
| Online VAR | Substitution | 680 | 8192 | 0.269 | 73.9% | 24.29 | 63.8 | 0.120 | **1.49** | 187.7 |
| Wasserstein VAR | Substitution | 680 | 8192 | 0.240 | **100%** | 24.69 | **65.1** | 0.112 | 1.55 | 190.5 |
| MMD VAR | Substitution | 680 | 8192 | **0.234** | **100%** | **24.73** | **65.1** | 0.111 | **1.49** | **190.4** |
| Vanilla VAR | Adaptation | 680 | 4096 | 0.305 | 38.2% | 23.22 | 60.1 | 0.126 | 1.25 | 185.6 |
| EMA VAR | Adaptation | 680 | 4096 | 0.321 | 99.9% | 22.90 | 59.1 | 0.139 | 1.69 | 177.4 |
| Online VAR | Adaptation | 680 | 4096 | 0.276 | 99.0% | 23.74 | 61.8 | 0.117 | 1.05 | 193.0 |
| Wasserstein VAR | Adaptation | 680 | 4096 | **0.255** | **100%** | 24.10 | 62.9 | 0.109 | 0.93 | 196.9 |
| MMD VAR | Adaptation | 680 | 4096 | **0.255** | **100%** | **24.16** | **63.2** | 0.108 | **0.91** | **199.2** |
| Vanilla VAR | Adaptation | 680 | 8192 | 0.309 | 22.9% | 23.02 | 60.1 | 0.128 | 1.30 | 185.5 |
| EMA VAR | Adaptation | 680 | 8192 | 0.312 | 99.8% | 23.02 | 59.3 | 0.137 | 1.15 | 191.9 |
| Online VAR | Adaptation | 680 | 8192 | 0.269 | 73.9% | 23.87 | 62.5 | 0.113 | 1.00 | 193.4 |
| Wasserstein VAR | Adaptation | 680 | 8192 | 0.240 | **100%** | **24.40** | **64.1** | **0.104** | 0.83 | 198.8 |
| MMD VAR | Adaptation | 680 | 8192 | **0.234** | **100%** | 24.37 | 63.8 | **0.104** | **0.81** | **201.0** |

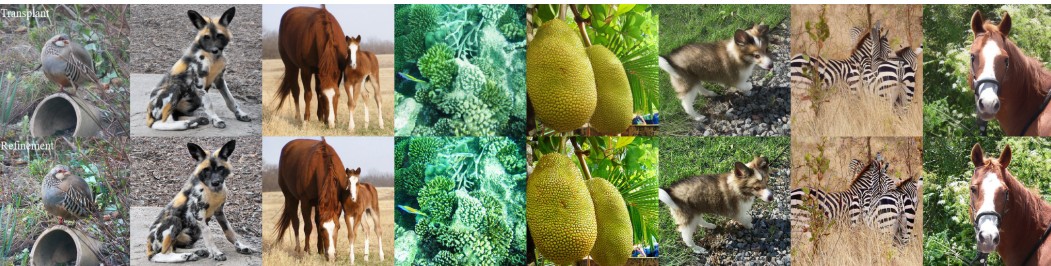

Figure 2: Comparison of reconstructed ImageNet-1k images based on MMD VAR: (Top) VQ Module Substitution Stage; (Bottom) Post-Decoder Adaptation Phase.

Table 4: r-FID progression during decoder adaptation on ImageNet-1K across epochs.

| Methods | Tokens | Codebook Size $K$ | Epoch 1 | Epoch 2 | Epoch 3 | Epoch 4 | Epoch 5 |
|---|---|---|---|---|---|---|---|
| Wasserstein VAR | 680 | 4096 | 1.006 | 0.968 | 0.957 | 0.953 | 0.926 |
| MMD VAR | 680 | 4096 | 1.023 | 0.987 | 0.950 | 0.937 | **0.911** |
| Wasserstein VAR | 680 | 8192 | 0.910 | 0.903 | 0.860 | 0.832 | 0.833 |
| MMD VAR | 680 | 8192 | 0.909 | 0.880 | 0.890 | 0.847 | **0.806** |

Table 5: Reconstruction performance of multi-scale VQ algorithms on ImageNet-1K dataset. The suffixes "–a", "–b", "–c", and "–d" correspond to decoder adaptation for 5, 10, 15, and 20 epochs, respectively. For each codebook size, the best-performing result is highlighted in bold.

| Methods | Phase | Tokens | Codebook Size $K$ | $\mathcal{E}(\downarrow)$ | $\mathcal{U}(\uparrow)$ | PSNR($\uparrow$) | SSIM($\uparrow$) | LPIPS ($\downarrow$) | r-FID($\downarrow$) | r-IS($\uparrow$) |
|---|---|---|---|---|---|---|---|---|---|---|
| VAR Tokenizer (Tian et al., 2024) | - | 680 | 4096 | 0.283 | 100% | **24.37** | **63.9** | **0.100** | 0.92 | 198.6 |
| MMD VAR-a | Adaptation | 680 | 4096 | **0.255** | 100% | 24.16 | 63.2 | 0.108 | 0.91 | 199.2 |
| MMD VAR-a | Adaptation | 680 | 8192 | **0.234** | 100% | 24.37 | 63.8 | 0.104 | 0.81 | **201.0** |
| MMD VAR-b | Adaptation | 680 | 4096 | **0.255** | 100% | 24.11 | 63.1 | 0.108 | 0.87 | **199.9** |
| MMD VAR-b | Adaptation | 680 | 8192 | **0.234** | 100% | 24.35 | **63.9** | **0.103** | 0.78 | **201.0** |
| MMD VAR-c | Adaptation | 680 | 4096 | **0.255** | 100% | 24.13 | 63.0 | 0.108 | 0.82 | 198.3 |
| MMD VAR-c | Adaptation | 680 | 8192 | **0.234** | 100% | **24.39** | 63.8 | **0.103** | 0.75 | **201.0** |
| MMD VAR-d | Adaptation | 680 | 4096 | **0.255** | 100% | 24.24 | 63.3 | 0.108 | **0.79** | 199.1 |
| MMD VAR-d | Adaptation | 680 | 8192 | **0.234** | 100% | 24.36 | 63.7 | **0.103** | 0.74 | **201.0** |

replacing the native VQ modules in a pretrained VAR tokenizer, MMD VAR demonstrates the highest level of compatibility. Wasserstein VAR follows closely, achieving nearly equivalent compatibility by achieving both **lower quantization error** and **100% codebook utilization**, outperforming other methods like Vanilla VAR, EMA VAR, and Online VAR. These results suggest that VQ algorithms based on distributional alignment minimize information loss during substitution, thereby enabling superior reconstruction performance compared to conventional approaches.

However, substituting the VQ module introduces misalignment between the features expected by the decoder and those produced by the new quantized latent space. Two key observations support this: First, while MMD VAR achieve lower quantization error than the original VAR tokenizer (0.255 and 0.234 vs. 0.283), their reconstruction metrics (r-FID and r-IS in Table 3) remain inferior to those of the original model (Tian et al., 2024). Second, Figure 2 reveals blurring and loss of high-frequency detail in reconstructions using the substituted VQ modules. These findings confirm a persistent **mismatch between the decoder's learned priors and the altered quantized latent space**.

To resolve this misalignment, we implemented lightweight decoder adaptation on ImageNet-1k, fine-tuning the decoder parameters for just 5 epochs. This adaptation aims to align the decoder's learned priors with the altered quantized space. After adaptation, both Wasserstein VAR and MMD VAR surpass the performance of the original VAR tokenizer on both r-FID and r-IS metrics (Table 3). Additionally, as Figure 2 demonstrates, the resulting reconstructions exhibit dramatically improved visual fidelity. Further reconstruction samples comparing performance after decoder adaptation are provided in Figure 8 (Appendix).

**Analyses On Adaptation Epochs.** Table 4 tracks the progression of r-FID metrics during decoder adaptation on ImageNet-1K across training epochs. We observe consistent r-FID improvements for both MMD VAR and Wasserstein VAR as the number of adaptation epochs increases. To examine whether MMD VAR could achieve further gains beyond 5 epochs, we extended the adaptation to 20 epochs. As reported in Table 5, MMD VAR with codebook sizes of 4096 and 8192 achieves improved r-FID scores of **0.79 and 0.74**, respectively, representing a further improvement over the epoch-5 baselines of 0.91 and 0.81.

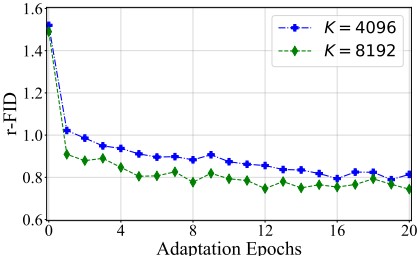

Figure 3: r-FID performance as a function of adaptation epochs.

Figure 3 shows the r-FID curves over 20 epochs. Despite some fluctuations, there is a clear overall downward trend, demonstrating that continued adversarial training effectively converts the reduction in quantization error achieved by MMD VAR in Stage I into improved reconstruction performance.

**Comparison Between From-scratch Training and VQ-Transplant.** We also compare from-scratch training with VQ-Transplant. As shown in Table 6, by examining the MMD VAR on ImageNet-1k, we find that even when from-scratch training is run for longer than VQ-Transplant, it still yields

Table 6: Reconstruction performance of multi-scale VQ algorithms on ImageNet-1K dataset. The suffixes "–a," "–b," and "–c" correspond to from-scratch training for 5, 6, and 7 epochs, respectively. For each codebook size, the best-performing result is highlighted in bold.

| Methods | Training Strategy | Training Hours | Tokens | Codebook Size $K$ | $\mathcal{E}(\downarrow)$ | $\mathcal{U}(\uparrow)$ | PSNR($\uparrow$) | SSIM($\uparrow$) | LPIPS ($\downarrow$) | r-FID($\downarrow$) | r-IS($\uparrow$) |
|---------|-------------------|----------------|--------|-------------------|-----|-----|------|------|-------|-------|-------|
| MMD VAR | VQ-Transplant | 22 | 680 | 4096 | 0.255 | **100%** | **24.16** | **63.2** | **0.108** | **0.91** | **199.2** |
| MMD VAR | VQ-Transplant | 22 | 680 | 8192 | 0.234 | **100%** | 24.37 | 63.8 | **0.104** | **0.81** | **201.0** |
| MMD VAR-a | From-scratch training | 25 | 680 | 4096 | 0.769 | **100%** | 23.63 | 60.7 | 0.120 | 1.40 | 189.4 |
| MMD VAR-a | From-scratch training | 25 | 680 | 8192 | 0.659 | **100%** | 23.58 | 60.6 | 0.120 | 1.26 | 191.0 |
| MMD VAR-b | From-scratch training | 30 | 680 | 4096 | 0.744 | **100%** | 23.74 | 61.1 | 0.120 | 1.36 | 191.9 |
| MMD VAR-b | From-scratch training | 30 | 680 | 8192 | 0.665 | **100%** | 23.70 | 60.9 | 0.119 | 1.29 | 190.2 |
| MMD VAR-c | From-scratch training | 35 | 680 | 4096 | 0.753 | **100%** | 23.74 | 61.1 | 0.119 | 1.34 | 190.4 |
| MMD VAR-c | From-scratch training | 35 | 680 | 8192 | 0.662 | **100%** | 23.75 | 60.9 | 0.119 | 1.26 | 191.0 |

relatively poor reconstruction performance. This outcome is expected, as discrete tokenizers typically require hundreds of epochs to achieve high-quality visual reconstruction when trained from scratch. In contrast, VQ-Transplant offers a more favorable trade-off between performance and training time, yielding accurate reconstruction while consuming significantly fewer computational resources.

**Joint Optimization of Encoder, Decoder, and VQ in Stage II** We also discuss an alternative optimization approach in Appendix C, where the encoder, decoder, and VQ module are jointly optimized, instead of optimizing only the decoder. As shown in Table 14, although one might worry that joint optimization could cause mode collapse, in practice it is stable, just like the decoder-only approach, and for most VQ methods it further improves reconstruction performance. However, these performance gains come at the cost of increased training time.

**Compatibility of VQ-Transplant with Pretrained LDM Tokenizer.** To evaluate the compatibility of VQ-Transplant with different tokenizers, we applied it to the pretrained LDM-16 continuous tokenizer (Rombach et al., 2022). As shown in Table 16, VQ-Transplant achieves reasonable performance on LDM-16. Nevertheless, its adaptability is lower compared to VAR-based models, as shown in Table 3, particularly with respect to r-FID and r-IS metrics. We provide a detailed discussion of this performance gap in Appendix D.

## 5.2 INTEGRATION OF FIXED-SCALE VECTOR QUANTIZATION ALGORITHMS

We further investigate integrating five fixed-scale VQ algorithms within the VQ-Transplant framework: Vanilla VQ, EMA VQ, Online VQ, Wasserstein VQ, and MMD VQ. As shown in Table 7, nearly identical performance patterns emerge to those observed in Table 3, demonstrating consistent transferability of conclusions from multi-scale to fixed-scale VQ algorithms. Three key observations follow: (1) Distribution-alignment VQ algorithms (e.g., Wasserstein VQ and MMD VQ) consistently achieve the lowest quantization error across all codebook sizes, demonstrating their superiority in minimizing information loss when replacing VQ modules. (2) Crucially, reduced quantization error does not translate to improved r-FID or r-IS metrics relative to the original VAR tokenizer (Tian et al., 2024), indicating a misalignment between the decoder's learned priors and the modified quantized latent space. (3) After realigning decoder priors with the altered latent space via decoder adaptation, all models substantially improve reconstruction fidelity, confirming the importance of decode adaptation in the VQ-Transplant framework. Representative reconstruction samples post-adaptation are shown in Figure 9 in the Appendix.

## 5.3 CROSS-DATASET GENERALIZATION OF VQ-TRANSPLANT

Finally, we evaluate the cross-dataset generalization capacity of the VQ-Transplant framework. While prior experiments demonstrate strong reconstruction fidelity on ImageNet-1k (Deng et al., 2009), its generalizability across diverse domains remains underexplored. This limitation arises because the original VAR tokenizer was trained on OpenImages (Kuznetsova et al., 2018)–where ImageNet-1k is a subset–raising a critical question: *Can the framework generalize to datasets structurally distinct from both ImageNet-1k and OpenImages?* To answer this, we empirically validate VQ-Transplant's cross-dataset performance on three divergent datasets: CelebA-HQ (Karras et al., 2017), FFHQ (Karras et al., 2019), and LSUN-Churches (Yu et al., 2015).

As detailed in Tables 8, 9, and 10, Wasserstein VQ and MMD VQ—implemented via the VQ-Transplant framework with fixed-scale quantization—demonstrate exceptional cross-dataset generalization, achieving state-of-the-art reconstruction performance across all three benchmarks. This generalization capacity is visually corroborated by high-fidelity samples in Figures 4, 5, and 6. Most

Table 7: Reconstruction performance of fixed-scale VQ algorithms on ImageNet-1K dataset. For each codebook size and each phase, the best-performing result is highlighted in bold.

| VQs | Phase | Tokens | Codebook Size $K$ | $\mathcal{E}(\downarrow)$ | $\mathcal{U}(\uparrow)$ | PSNR($\uparrow$) | SSIM($\uparrow$) | LPIPS ($\downarrow$) | r-FID($\downarrow$) | r-IS($\uparrow$) |
|---|---|---|---|---|---|---|---|---|---|---|
| Vanilla VQ | Substitution | 512 | 16384 | 0.423 | 0.8% | 22.03 | 53.0 | 0.244 | 10.61 | 104.7 |
| EMA VQ | Substitution | 512 | 16384 | 0.240 | **100%** | 24.71 | 64.8 | 0.134 | 1.86 | 181.2 |
| Online VQ | Substitution | 512 | 16384 | 0.286 | 42.0% | 24.32 | 62.7 | 0.149 | 2.20 | 174.1 |
| Wasserstein VQ | Substitution | 512 | 16384 | **0.231** | 99.8% | 24.84 | **65.4** | **0.130** | **1.69** | **184.8** |
| MMD VQ | Substitution | 512 | 16384 | 0.234 | 99.8% | **24.89** | **65.4** | **0.130** | 1.84 | 183.7 |
| Vanilla VQ | Substitution | 512 | 32768 | 0.422 | 0.5% | 22.02 | 53.1 | 0.242 | 10.53 | 106.2 |
| EMA VQ | Substitution | 512 | 32768 | 0.223 | **100%** | 24.89 | 65.7 | 0.129 | 1.81 | 183.0 |
| Online VQ | Substitution | 512 | 32768 | 0.279 | 22.7% | 24.43 | 63.1 | 0.146 | 2.13 | 175.6 |
| Wasserstein VQ | Substitution | 512 | 32768 | **0.215** | 99.7% | **25.06** | **66.3** | 0.126 | 1.79 | 184.8 |
| MMD VQ | Substitution | 512 | 32768 | 0.216 | 99.9% | 25.04 | 66.2 | **0.124** | **1.73** | **187.0** |
| Vanilla VQ | Substitution | 512 | 65536 | 0.422 | 0.2% | 22.04 | 53.1 | 0.243 | 10.89 | 103.8 |
| EMA VQ | Substitution | 512 | 65536 | 0.217 | 65.5% | 24.94 | 65.9 | 0.127 | 1.78 | 185.8 |
| Online VQ | Substitution | 512 | 65536 | 0.280 | 13.5% | 24.42 | 63.2 | 0.147 | 2.28 | 174.2 |
| Wasserstein VQ | Substitution | 512 | 65536 | **0.201** | 99.6% | 25.22 | **66.9** | **0.121** | 1.76 | 186.0 |
| MMD VQ | Substitution | 512 | 65536 | **0.201** | **99.9%** | **25.24** | 66.8 | **0.121** | **1.69** | **187.3** |
| Vanilla VQ | Adaptation | 512 | 16384 | 0.423 | 0.8% | 21.36 | 51.3 | 0.208 | 5.02 | 118.4 |
| EMA VQ | Adaptation | 512 | 16384 | 0.240 | **100%** | 24.12 | 63.2 | 0.118 | 1.11 | 190.0 |
| Online VQ | Adaptation | 512 | 16384 | 0.286 | 42.0% | 23.78 | 60.9 | 0.132 | 1.49 | 178.9 |
| Wasserstein VQ | Adaptation | 512 | 16384 | **0.231** | 99.8% | **24.36** | **64.0** | **0.114** | **1.04** | **191.3** |
| MMD VQ | Adaptation | 512 | 16384 | 0.234 | 99.8% | 24.31 | 63.7 | 0.115 | 1.05 | 191.2 |
| Vanilla VQ | Adaptation | 512 | 32768 | 0.422 | 0.5% | 21.22 | 51.0 | 0.209 | 5.11 | 117.4 |
| EMA VQ | Adaptation | 512 | 32768 | 0.223 | **100%** | 24.24 | 63.6 | 0.113 | 0.99 | 192.2 |
| Online VQ | Adaptation | 512 | 32768 | 0.279 | 22.7% | 23.86 | 61.6 | 0.129 | 1.40 | 181.3 |
| Wasserstein VQ | Adaptation | 512 | 32768 | **0.215** | 99.7% | 24.37 | 64.3 | 0.111 | 0.98 | 193.9 |
| MMD VQ | Adaptation | 512 | 32768 | 0.216 | 99.9% | **24.53** | **64.7** | **0.110** | **0.97** | **194.1** |
| Vanilla VQ | Adaptation | 512 | 65536 | 0.422 | 0.2% | 21.19 | 50.7 | 0.209 | 5.05 | 118.9 |
| EMA VQ | Adaptation | 512 | 65536 | 0.217 | 65.5% | 24.36 | 64.1 | 0.111 | 0.99 | 194.3 |
| Online VQ | Adaptation | 512 | 65536 | 0.280 | 13.5% | 23.84 | 61.6 | 0.130 | 1.38 | 182.9 |
| Wasserstein VQ | Adaptation | 512 | 65536 | **0.201** | 99.6% | **24.68** | **65.4** | **0.106** | 0.92 | 195.5 |
| MMD VQ | Adaptation | 512 | 65536 | **0.201** | **99.9%** | 24.65 | 65.0 | **0.106** | **0.86** | **197.1** |

Table 8: Reconstruction performance on the FFHQ dataset. [†]: results cited from (Zhu et al., 2024).

| VQs | Phase | Tokens | Codebook Size $K$ | $\mathcal{E}(\downarrow)$ | $\mathcal{U}(\uparrow)$ | PSNR($\uparrow$) | SSIM($\uparrow$) | LPIPS ($\downarrow$) | r-FID($\downarrow$) |
|---|---|---|---|---|---|---|---|---|---|
| RQVAE[†] | Full Training | 256 | 2048 | - | - | 22.9 | 67.0 | 0.13 | 7.04 |
| VQ-WAE[†] | Full Training | 256 | 1024 | - | - | 22.5 | 66.5 | 0.12 | 4.20 |
| MQVAE[†] | Full Training | 256 | 1024 | - | 78.2% | - | - | - | 4.55 |
| VQGAN[†] | Full Training | 256 | 16384 | - | 2.3% | 24.4 | 63.3 | 0.12 | 5.25 |
| VQGAN-FC[†] | Full Training | 256 | 16384 | - | 10.9% | 24.8 | 64.6 | 0.11 | 4.86 |
| VQGAN-EMA[†] | Full Training | 256 | 16384 | - | 68.2% | 25.4 | 66.1 | 0.10 | 4.79 |
| VQGAN-LC[†] | Full Training | 256 | 100000 | - | 99.5% | 26.1 | 69.4 | 0.08 | 3.81 |
| Wasserstein VQ | Substitution | 512 | 16384 | **0.153** | 99.7% | 27.59 | **77.2** | 0.076 | 2.63 |
| MMD VQ | Substitution | 512 | 16384 | **0.153** | **99.9%** | 27.63 | 77.0 | **0.075** | **2.21** |
| Wasserstein VQ | Substitution | 512 | 32768 | **0.142** | 99.7% | 27.83 | **77.7** | **0.072** | 2.27 |
| MMD VQ | Substitution | 512 | 32768 | **0.142** | **99.9%** | **27.88** | **77.7** | 0.073 | **2.09** |
| Wasserstein VQ | Adaptation | 512 | 16384 | **0.153** | 99.7% | **27.25** | **75.4** | **0.075** | **1.81** |
| MMD VQ | Adaptation | 512 | 16384 | **0.153** | **99.9%** | 27.09 | 74.7 | **0.075** | 1.99 |
| Wasserstein VQ | Adaptation | 512 | 32768 | **0.142** | 99.7% | 27.33 | **75.7** | **0.072** | **1.21** |
| MMD VQ | Adaptation | 512 | 32768 | **0.142** | **99.9%** | **27.43** | 75.5 | 0.073 | 1.37 |

Table 9: Reconstruction performance on the CelebA-HQ dataset. For each codebook size and each phase, the best-performing result is highlighted in bold.

| VQs | Phase | Tokens | Codebook Size $K$ | $\mathcal{E}(\downarrow)$ | $\mathcal{U}(\uparrow)$ | PSNR($\uparrow$) | SSIM($\uparrow$) | LPIPS ($\downarrow$) | r-FID($\downarrow$) |
|---|---|---|---|---|---|---|---|---|---|
| Wasserstein VQ | Substitution | 512 | 16384 | **0.129** | 99.5% | 27.92 | **77.6** | 0.070 | 3.30 |
| MMD VQ | Substitution | 512 | 16384 | **0.129** | **99.6%** | **28.02** | 77.5 | **0.069** | **2.96** |
| Wasserstein VQ | Adaptation | 512 | 16384 | **0.129** | 99.5% | 27.45 | 75.2 | **0.071** | 3.02 |
| MMD VQ | Adaptation | 512 | 16384 | **0.129** | **99.6%** | **27.71** | **75.7** | 0.071 | **2.60** |

Table 10: Reconstruction performance on the Churches dataset. For each codebook size and each phase, the best-performing result is highlighted in bold.

| VQs | Phase | Tokens | Codebook Size $K$ | $\mathcal{E}(\downarrow)$ | $\mathcal{U}(\uparrow)$ | PSNR($\uparrow$) | SSIM($\uparrow$) | LPIPS ($\downarrow$) | r-FID($\downarrow$) |
|---|---|---|---|---|---|---|---|---|---|
| Wasserstein VQ | Substitution | 512 | 16384 | **0.204** | 99.6% | **23.35** | **64.2** | 0.144 | 2.76 |
| MMD VQ | Substitution | 512 | 16384 | **0.204** | **99.7%** | 23.34 | **64.2** | **0.141** | **2.72** |
| Wasserstein VQ | Adaptation | 512 | 16384 | **0.204** | 99.6% | **22.63** | **62.3** | **0.122** | **1.79** |
| MMD VQ | Adaptation | 512 | 16384 | **0.204** | **99.7%** | 22.51 | 61.7 | 0.124 | 1.87 |

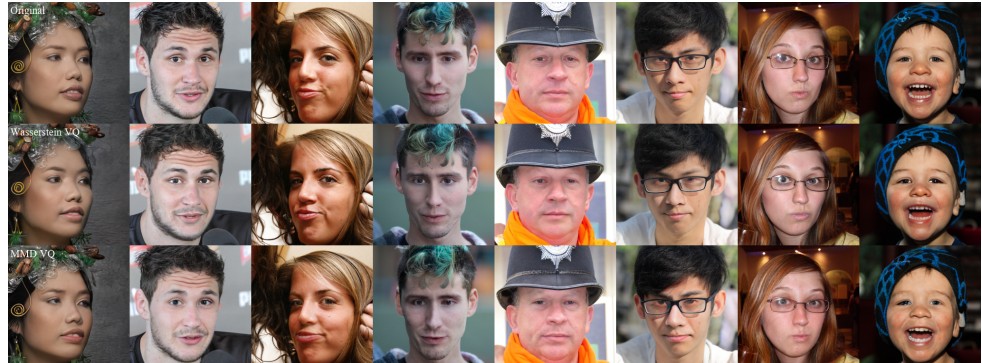

Figure 4: Reconstructed FFHQ Samples. (Top) Original inputs; (Middle) Wasserstein VQ reconstruction; (Bottom) MMD VQ reconstruction. All images are $256 \times 256$ resolution.

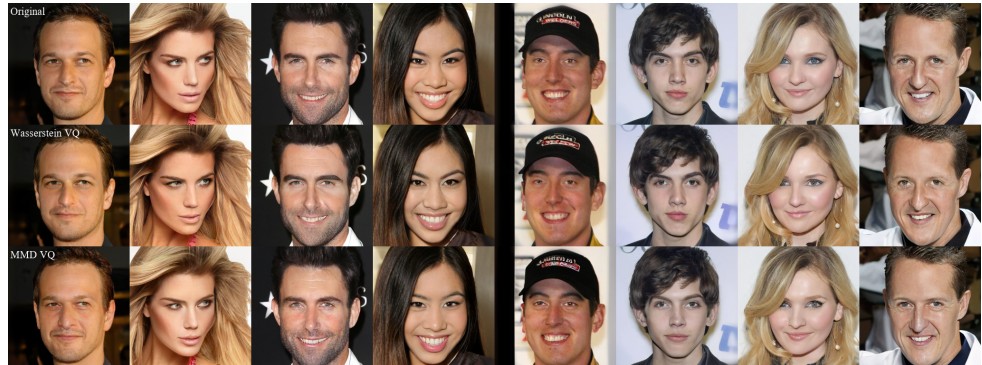

Figure 5: Reconstructed CelebA-HQ Samples. (Top) Original inputs; (Middle) Wasserstein VQ reconstruction; (Bottom) MMD VQ reconstruction. All images are $256 \times 256$ resolution.

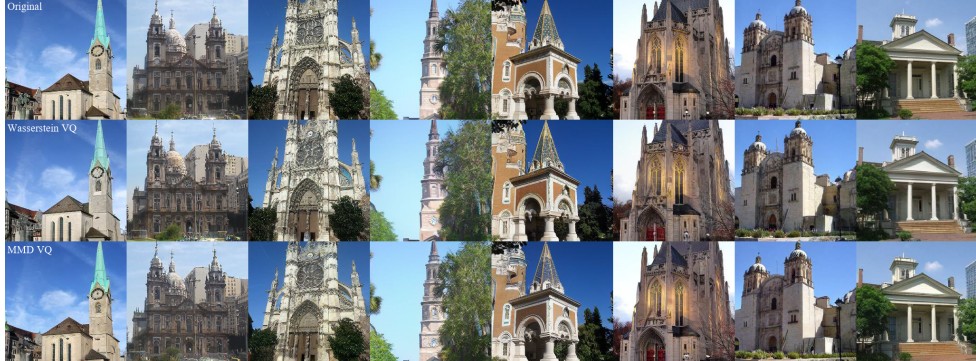

Figure 6: Reconstructed LSUN-Churches Samples. (Top) Original inputs; (Middle) Wasserstein VQ reconstruction; (Bottom) MMD VQ reconstruction. All images are $256 \times 256$ resolution.

notably on FFHQ (Table 8), Wasserstein VQ achieves a record r-FID of 1.21, significantly outperforming all baselines including RQVAE (Lee et al., 2022), VQGAN (Esser et al., 2021), VQGAN-FC (Yu et al., 2022), VQGAN-EMA (Razavi et al., 2019), VQ-WAE (Vuong et al., 2023), MQVAE (Huang et al., 2023b), and VQGAN-LC (Zhu et al., 2024).

## 6 CONCLUSION

In this paper, we proposed **VQ-Transplant**, a computationally efficient framework that enables rapid plug-and-play integration of novel VQ algorithms into pre-trained visual tokenizers—eliminating costly retraining requirements—as well as **MMD-VQ**, a quantization method that uses maximum mean discrepancy to force distributional alignment and ensure compatibility with VQ-Transplant. We hope that this work will help democratize quantization research by enabling resource-efficient integration of novel VQ techniques while matching industry-level reconstruction performance.

## 7 REPRODUCIBILITY STATEMENT

To ensure full reproducibility of our results, the following resources are included in the supplemental materials: (1) complete training and evaluation source code, (2) execution scripts for all experiments, (3) comprehensive training logs capturing model dynamics, and (4) final model outputs and evaluation artifacts. To further support the research community, all resources—including pre-trained model weights, detailed documentation, and configuration files—will be publicly released on GitHub. This release will enable independent verification of our findings and facilitate future research.

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

# APPENDIX

## A    EXPERIMENTAL DETAILS

**Data Augmentation.**    For all four datasets—ImageNet-1k (Deng et al., 2009), FFHQ (Karras et al., 2019), CelebA-HQ (Karras et al., 2017), and LSUN-Churches (Yu et al., 2015)—we follow Llama Gen (Sun et al., 2024) by applying iterative box downsampling to resize all images to a 256×256 resolution.

**Encoder-Decoder Architecture.**    For all experiments within the VQ-Transplant framework, we maintain the identical encoder-decoder architecture used in the VAR tokenizer (Tian et al., 2024). The encoder—a U-Net (Ronneberger et al., 2015)—downsamples input images by a factor of 16, resulting in latent features with $16 \times 16$ spatial resolution.

**Training Details.**    All models were trained on two NVIDIA H100 GPUs using the AdamW optimizer (Loshchilov & Hutter, 2019) with $\beta_1 = 0.9$ and $\beta_1 = 0.95$. During VQ module substitution, we used an initial learning rate of $10^{-4}$ with linear decay to $10^{-5}$. For decoder adaptation, the learning rate remained constant at $10^{-5}$. The number of training epochs for each experiment configuration is shown in Table 11. The batch size for all experiments was 32 per GPU.

Table 11: Training epochs across various dataset.

| Datasets | ImageNet-1k | FFHQ | CelebA-HQ | LSUN-Churches |
|---|---|---|---|---|
| Substitution Epochs | 2 | 30 | 30 | 20 |
| Adaptation Epochs | 5 | 30 | 30 | 20 |

**Loss Weight.**    For all experiments, $\lambda_P$ is fixed to 1. In multi-scale quantization experiments, $\lambda_G = 0.5$, while in fixed-scale quantization experiments, $\lambda_G = 0.4$. We set $\gamma = 0.2$ for configurations employing Wasserstein distance (Wasserstein VAR and VQ) and $\gamma = 0.5$ for for configurations using MMD distance (MMD VAR and VQ).

## B    THE MOTIVATION OF MMD VQ: FROM WASSERSTEIN DISTANCE TO MMD DISTANCE

The motivation for using Maximum Mean Discrepancy (MMD) arises from a well-established theoretical understanding of vector quantization (VQ): *VQ becomes near-optimal when the codebook distribution matches the underlying feature distribution, yielding both minimal quantization error and maximal codebook utilization.* This principle is rigorously supported in prior work (Fang et al., 2025; Graf & Luschgy, 2000). Motivated by this insight, we aim to approximate the optimality condition during training by explicitly encouraging alignment between the encoder feature distribution and the learned codebook distribution. MMD provides an effective mechanism for this alignment through its use of characteristic kernels that guarantee universal distribution matching.

**Wasserstein Distance: Definition and Context.**    To motivate the choice of MMD, we first situate it in relation to the Wasserstein distance (WD), a metric used in prior VQ research for distribution alignment (Fang et al., 2025). The Wasserstein distance provides a geometric notion of discrepancy by measuring the minimal cost required to transport mass from one distribution to another. Formally, the $p$-Wasserstein distance is defined as:

$$W_p(\mathbb{P}_r, \mathbb{P}_g) = (\inf_{\gamma \in \Pi(\mathbb{P}_r, \mathbb{P}_g)} \mathbb{E}_{(x,y)\sim\gamma}\big[\|x - y\|^p\big])^{1/p} . \tag{6}$$

where $\Pi(\mathbb{P}_r, \mathbb{P}_g)$ denotes the set of all couplings whose marginals are $\mathbb{P}_r$ and $\mathbb{P}_g$. Intuitively, Wasserstein distance—also known as the earth-mover distance—measures the minimum amount of "work" needed to transform $\mathbb{P}_r$ into $\mathbb{P}_g$. The case $p = 2$ corresponds to the quadratic Wasserstein distance.

Table 12: Quantization error under varying levels of non-Gaussianity..

| Methods | $\zeta = 0.0$ | $\zeta = 0.5$ | $\zeta = 1.0$ | $\zeta = 1.5$ | $\zeta = 2.0$ | $\zeta = 2.5$ | $\zeta = 3.0$ | $\zeta = 3.5$ | $\zeta = 4.0$ |
|---|---|---|---|---|---|---|---|---|---|
| Wasserstein VQ | 0.976 | 1.099 | 1.177 | 1.252 | 1.318 | 1.373 | 1.420 | 1.462 | 1.502 |
| MMD VQ | 0.968 | 1.088 | 1.142 | 1.155 | 1.171 | 1.186 | 1.198 | 1.217 | 1.240 |

Table 13: Codebook utilization under varying levels of non-Gaussianity.

| Methods | $\zeta = 0.0$ | $\zeta = 0.5$ | $\zeta = 1.0$ | $\zeta = 1.5$ | $\zeta = 2.0$ | $\zeta = 2.5$ | $\zeta = 3.0$ | $\zeta = 3.5$ | $\zeta = 4.0$ |
|---|---|---|---|---|---|---|---|---|---|
| Wasserstein VQ | 99.9% | 99.8% | 97.0% | 78.2% | 62.7% | 52.1% | 44.8% | 39.2% | 34.8% |
| MMD VQ | 99.9% | 99.9% | 99.8% | 97.0% | 92.5% | 88.9% | 85.7% | 81.2% | 75.6% |

| (a) MMD VQ | (b) Wasserstein VQ | (c) MMD VQ | (d) Wasserstein VQ |

Figure 7: Comparison between MMD-VQ and Wasserstein-VQ on non-Gaussian data.

**Limitation of Wasserstein Distance in VQ.** A well-known practical limitation of Wasserstein distance is that it is computationally intractable when applied directly during neural network training. To address this, Wasserstein VQ (Fang et al., 2025) assumes that both the feature distribution and the codebook distribution follow Gaussian forms,

$$\mathbb{P}_g = \mathcal{N}(\mu_1, \Sigma_1), \mathbb{P}_r = \mathcal{N}(\mu_2, \Sigma_2), \tag{7}$$

Under this Gaussianity assumption, the intractable optimization in Equation 6 reduces to a differentiable closed-form expression involving only means and covariances:

$$W_2(\mathcal{N}(\mu_1, \Sigma_1), \mathcal{N}(\mu_2, \Sigma_2)) \sqrt{\|\mu_1 - \mu_2\|_2^2 + \text{tr}\left(\Sigma_1 + \Sigma_2 - 2(\Sigma_1^{\frac{1}{2}} \Sigma_2 \Sigma_1^{\frac{1}{2}})^{\frac{1}{2}}\right)}. \tag{8}$$

However, this simplification has an important implication: **Wasserstein VQ aligns only first- and second-order moments** of the distributions. While sufficient when encoder features are approximately Gaussian, this restriction becomes limiting for the multi-modal, highly non-Gaussian, or heavy-tailed feature distributions commonly encountered in deep visual representations.

**Advantage of MMD-VQ.** In contrast, MMD leverages reproducing kernel Hilbert space (RKHS) embeddings with characteristic kernels (e.g., RBF), which guarantee that minimizing MMD corresponds to matching **all moments** of the distributions. This yields a substantially more expressive alignment mechanism and makes MMD-VQ inherently robust in scenarios where Gaussian assumptions do not hold.

**Observed Similar Empirical Performance** In standard benchmarks (ImageNet, FFHQ, CelebA-HQ, LSUN-Churches), encoder-produced latent features are typically approximately Gaussian, likely due to the aggregation of many weakly dependent components in deep representations, a phenomenon broadly consistent with the Central Limit Theorem (CLT). In such cases, aligning first and second moments is sufficient, and the advantage of MMD-VQ is less pronounced.

**Analyses on Non-Gaussian Data.** To investigate the practical impact of the moment-matching limitations of Wasserstein VQ, we conduct controlled experiments on synthetic latent feature distributions with varying degrees of non-Gaussianity. Specifically, we construct a bimodal mixture distribution:

$$\mathbb{P}_g = 0.5 \cdot \mathcal{N}(\zeta \mathbf{1}, I) + 0.5 \cdot \mathcal{N}(-\zeta \mathbf{1}, I), \tag{9}$$

where $\zeta \in \{0.0, 0.5, \ldots, 4.0\}$ controls the separation between the two modes. When $\zeta = 0$, the distribution reduces to a standard Gaussian; as $\zeta$ increases, it becomes progressively multi-modal and strongly non-Gaussian.

The codebook distribution $\mathbb{P}_r$ is initialized as a standard Gaussian. We set the codebook size to 16,384 with code vectors of dimension 8 and train both Wasserstein-VQ and MMD-VQ for 10,000 steps. At each step, we sample 20,000 feature vectors from $\mathbb{P}_g$, and record the quantization error and

Table 14: Reconstruction performance of multi-scale VQ algorithms on ImageNet-1K dataset. For each training strategy and codebook size, the best-performing result is highlighted in bold.

| Methods | Training Strategy | Tokens | Codebook Size $K$ | $\mathcal{E}(\downarrow)$ | $\mathcal{U}(\uparrow)$ | PSNR($\uparrow$) | SSIM($\uparrow$) | LPIPS ($\downarrow$) | r-FID($\downarrow$) | r-IS($\uparrow$) |
|---|---|---|---|---|---|---|---|---|---|---|
| Vanilla VAR | Decoder-only | 680 | 4096 | 0.305 | 38.2% | 23.22 | 60.1 | 0.126 | 1.25 | 185.6 |
| EMA VAR | Decoder-only | 680 | 4096 | 0.321 | 99.9% | 22.90 | 59.1 | 0.139 | 1.69 | 177.4 |
| Online VAR | Decoder-only | 680 | 4096 | 0.276 | 99.0% | 23.74 | 61.8 | 0.117 | 1.05 | 193.0 |
| Wasserstein VAR | Decoder-only | 680 | 4096 | **0.255** | **100%** | 24.10 | 62.9 | 0.109 | 0.93 | 196.9 |
| MMD VAR | Decoder-only | 680 | 4096 | **0.255** | **100%** | **24.16** | **63.2** | **0.108** | **0.91** | **199.2** |
| Vanilla VAR | Decoder-only | 680 | 8192 | 0.309 | 22.9% | 23.02 | 60.1 | 0.128 | 1.30 | 185.5 |
| EMA VAR | Decoder-only | 680 | 8192 | 0.312 | 99.8% | 23.02 | 59.3 | 0.137 | 1.15 | 191.9 |
| Online VAR | Decoder-only | 680 | 8192 | 0.269 | 73.9% | 23.87 | 62.5 | 0.113 | 1.00 | 193.4 |
| Wasserstein VAR | Decoder-only | 680 | 8192 | 0.240 | **100%** | **24.40** | **64.1** | **0.104** | 0.83 | 198.8 |
| MMD VAR | Decoder-only | 680 | 8192 | **0.234** | **100%** | 24.37 | 63.8 | **0.104** | **0.81** | **201.0** |
| Vanilla VAR | Joint Optimization | 680 | 4096 | 0.403 | 26.4% | 23.64 | 60.5 | 0.118 | 1.07 | 191.7 |
| EMA VAR | Joint Optimization | 680 | 4096 | 1.512 | 99.9% | 22.36 | 55.5 | 0.154 | 1.94 | 172.5 |
| Online VAR | Joint Optimization | 680 | 4096 | 0.297 | **100%** | 23.90 | 62.3 | 0.112 | 0.98 | 196.1 |
| Wasserstein VAR | Joint Optimization | 680 | 4096 | 0.273 | **100%** | 24.25 | 63.4 | 0.107 | **0.87** | 198.7 |
| MMD VAR | Joint Optimization | 680 | 4096 | 0.264 | **100%** | **24.35** | **63.6** | **0.106** | **0.87** | **199.5** |
| Vanilla VAR | Joint Optimization | 680 | 8192 | 0.416 | 12.1% | 23.59 | 60.4 | 0.119 | 1.09 | 190.7 |
| EMA VAR | Joint Optimization | 680 | 8192 | 1.384 | 99.9% | 22.31 | 55.4 | 0.152 | 1.95 | 169.3 |
| Online VAR | Joint Optimization | 680 | 8192 | 0.262 | 85.4% | 24.09 | 62.7 | 0.109 | 0.92 | 197.3 |
| Wasserstein VAR | Joint Optimization | 680 | 8192 | 0.229 | **100%** | 24.48 | 64.2 | **0.102** | 0.81 | **201.7** |
| MMD VAR | Joint Optimization | 680 | 8192 | 0.227 | **100%** | **24.51** | **64.4** | **0.102** | **0.79** | 201.0 |

codebook utilization every 100 steps. The evolution of these metrics throughout training is shown in Figure 7.

We evaluate the performance of both methods across different levels of non-Gaussianity, as summarized in Tables 12 and 13. For small $\zeta$, where the latent distribution remains approximately Gaussian, Wasserstein-VQ and MMD-VQ perform similarly, consistent with the observation that matching only the first two moments is sufficient in this regime. As $\zeta$ increases and the distribution becomes more strongly bimodal, the difference between the two methods becomes pronounced. Wasserstein-VQ experiences significant degradation in both quantization error and codebook utilization, reflecting its inability to capture higher-order structures beyond the first two moments. In contrast, MMD-VQ maintains robust performance, leveraging characteristic kernels to align all moments of the distributions.

These experiments empirically confirm the theoretical expectations discussed earlier: MMD-VQ provides a clear advantage whenever the latent feature distribution deviates from Gaussianity, while Wasserstein-VQ remains competitive only when the Gaussian assumption approximately holds. Therefore, MMD-VQ demonstrates superior generalization capability: in future research scenarios where the latent feature distribution deviates from Gaussianity, MMD-VQ is expected to provide a more robust and effective alignment, offering clear advantages over Wasserstein-VQ.

## C  STAGE II ALTERNATIVE: AN JOINT OPTIMIZATION OF ENCODER, DECODDR, AND VQ

In the original Stage II in section 4.1, only the decoder $\mathcal{D}_\varphi$ is updated while keeping the pretrained encoder $\mathcal{E}_{\theta*}$ and newly trained VQ module $\mathcal{Q}_{\phi^\star}^{\text{new}}$ frozen. This approach addresses the mismatch between the updated quantized latent space and the frozen decoder, but it does not allow the encoder to adapt to the new quantization or jointly refine the reconstruction capability.

As an alternative, we also introduce a joint optimization scheme in Stage II, where the encoder, decoder, and VQ module are updated simultaneously. Let $z_e = \mathcal{E}_\theta(x)$ denote the encoder's latent embedding, and $z_q = \mathcal{Q}_\phi(ze)$ denote the quantized latent from the VQ module. The decoder reconstructs the input as $\widehat{x} = \mathcal{D}_\varphi(z_q)$. The overall joint optimization objective integrates both the VQ reconstruction loss and the decoder reconstruction loss:

$$\mathcal{L}_{\text{Joint}}(\theta, \phi, \varphi) = \|\text{sg}(z_e) - z_q\|_2^2 + \beta\|z_e - \text{sg}(z_q)\|_2^2 + \gamma\mathcal{L}_{\text{unique}}(\mathcal{Q}_\phi^{\text{new}}),$$
$$+ \|\widehat{x} - x\|_2^2 + \lambda_P\mathcal{L}_{\text{Per}} + \lambda_G\mathcal{L}_{\text{GAN}},$$

where $\mathcal{L}_{\text{unique}}$ enforces codebook uniqueness (e.g., Wasserstein loss for Wasserstein VQ (Fang et al., 2025) or MMD loss for MMD VQ), and $\mathcal{L}_{\text{Per}}$ and $\mathcal{L}_{\text{GAN}}$ correspond to perceptual and adversarial losses that promote high-quality reconstruction. The parameter $\beta$ is fixed to 0.25, while $\gamma$, $\lambda_P$, and $\lambda_G$ are hyperparameters balancing the respective terms, as detailed in Appendix A.

Table 15: Training time comparison between decoder-only and joint optimization strategies.

| Strategies | Stage I Epochs | Hours Per Epoch | Stage II Epochs | Hours Per Epoch | Total Hours |
|---|---|---|---|---|---|
| Decoder-Only | 2 | 2.25 | 5 | 3.5 | 22 |
| Joint Optimization | 2 | 2.25 | 5 | 5 | 29.5 |

In this setup, all three components—encoder $\mathcal{E}_\theta$, decoder $\mathcal{D}_\varphi$, and VQ module $\mathcal{Q}_\phi$, are updated jointly. To initialize the training, we load all parameters from Stage I, ensuring that the encoder, decoder, and VQ module start from the previously optimized representations. Joint optimization enables the encoder to adapt to the updated quantized space, allows the VQ module to refine the codebook representations, and improves the decoder's ability to reconstruct images accurately from the newly quantized latent features. For adversarial training, we follow prior works (Tian et al., 2024; Chen et al., 2025a; Li et al., 2025) and employ a frozen DINO-S (Caron et al., 2021; Oquab et al., 2024) discriminator with a StyleGAN-like architecture (Karras et al., 2020; 2019), augmented with DiffAug (Zhao et al., 2020), consistency regularization (Zhang et al., 2019), and LeCAM regularization (Tseng et al., 2021).

We conduct experiments on ImageNet-1K to compare the decoder-only and joint optimization strategies, as summarized in Table 14. Joint optimization maintains full codebook utilization (100%) for MMD-VAR, and thus does not experience the mode collapse that the reviewer had anticipated, while providing slight improvements across most metrics, with the exception of EMA-VAR. However, as shown in Table 15, this approach entails additional training time. Thus, while joint optimization offers slightly stronger performance, we chose decoder-only adaptation in the main submission to emphasize efficiency, a core design goal of VQ-Transplant.

## D    COMPATIBILITY OF VQ-TRANSPLANT WITH PRETRAINED LDM TOKENIZER

To evaluate the compatibility of VQ-Transplant with different tokenizers, we applied it to the pretrained LDM-16 continuous tokenizer (Rombach et al., 2022). As shown in Table 16, VQ-Transplant achieves reasonable performance on LDM-16. Nevertheless, its adaptability is lower compared to VAR-based models, as compared in Table 3, particularly in terms of r-FID and r-IS metrics.

We identify two primary factors that likely contribute to this performance gap:

- **Model capacity differences:** The VAR tokenizer employs a larger encoder–decoder architecture (104M parameters) compared to LDM-16 (68M), providing stronger reconstruction capabilities and greater flexibility during adaptation.

- **Decoder adaptation behavior:** The VAR decoder is pretrained on quantized features $z_q$, making it inherently robust to the perturbations introduced by new VQ modules. In contrast, the LDM decoder is trained exclusively on continuous features $z_e$, which limits its ability to adapt to discrete VQ representations. As a result, VAR-based VQ-Transplant achieves strong performance with only a few adaptation epochs, whereas LDM-based variants may require more extensive training.

These observations indicate that while VQ, Transplant is generally compatible with different tokenizers, the intrinsic properties of the base model—such as encoder–decoder capacity and the nature of pretrained features, play a critical role in determining adaptation efficiency.

Table 16: Reconstruction performance of multi-scale VQ algorithms on ImageNet-1K dataset based on LDM-16 Tokenizer. For each codebook size and each phase, the best-performing result is highlighted in bold

| Methods | Phase | Tokens | Codebook Size $K$ | $\mathcal{E}(\downarrow)$ | $\mathcal{U}(\uparrow)$ | PSNR(↑) | SSIM(↑) | LPIPS (↓) | r-FID(↓) | r-IS(↑) |
|---|---|---|---|---|---|---|---|---|---|---|
| LDM-16 (Rombach et al., 2022) | - | 256 | - | 0.0 | - | 24.08 | 68.0 | - | 0.87 | 210.3 |
| Vanilla VAR | Substitution | 680 | 4096 | 0.424 | 97.0% | 22.89 | 55.5 | 0.165 | 9.74 | 120.7 |
| EMA VAR | Substitution | 680 | 4096 | 0.367 | **100%** | 23.24 | 57.0 | 0.153 | 7.71 | 134.4 |
| Online VAR | Substitution | 680 | 4096 | 0.299 | **100%** | 23.65 | 58.7 | 0.139 | 5.62 | 150.3 |
| Wasserstein VAR | Substitution | 680 | 4096 | **0.278** | **100%** | **23.68** | **59.1** | **0.136** | **5.17** | **153.5** |
| MMD VAR | Substitution | 680 | 4096 | **0.278** | **100%** | 23.64 | 59.0 | **0.136** | 5.18 | 153.3 |
| Vanilla VAR | Substitution | 680 | 8192 | 0.418 | 87.3% | 22.86 | 55.4 | 0.167 | 9.83 | 120.4 |
| EMA VAR | Substitution | 680 | 8192 | 0.333 | **100%** | 23.40 | 57.8 | 0.146 | 6.63 | 142.2 |
| Online VAR | Substitution | 680 | 8192 | 0.283 | 94.7% | 23.76 | 59.1 | 0.135 | 5.14 | 153.4 |
| Wasserstein VAR | Substitution | 680 | 8192 | **0.252** | **100%** | **23.84** | **59.8** | **0.131** | **4.49** | **159.8** |
| MMD VAR | Substitution | 680 | 8192 | 0.254 | **100%** | 23.80 | 59.7 | **0.131** | 4.59 | 158.6 |
| Vanilla VAR | Adaptation | 680 | 4096 | 0.424 | 97.0% | 22.66 | 56.0 | 0.157 | 4.89 | 136.7 |
| EMA VAR | Adaptation | 680 | 4096 | 0.367 | **100%** | 22.88 | 57.0 | 0.147 | 4.11 | 144.9 |
| Online VAR | Adaptation | 680 | 4096 | 0.299 | **100%** | 23.45 | 59.1 | 0.134 | 3.18 | 158.8 |
| Wasserstein VAR | Adaptation | 680 | 4096 | **0.278** | **100%** | 23.49 | **59.5** | **0.130** | **2.87** | **161.4** |
| MMD VAR | Adaptation | 680 | 4096 | **0.278** | **100%** | **23.51** | **59.5** | 0.131 | 2.93 | 161.2 |
| Vanilla VAR | Adaptation | 680 | 8192 | 0.418 | 87.3% | 22.58 | 56.0 | 0.157 | 5.14 | 132.6 |
| EMA VAR | Adaptation | 680 | 8192 | 0.333 | **100%** | 23.26 | 58.4 | 0.139 | 3.59 | 151.6 |
| Online VAR | Adaptation | 680 | 8192 | 0.283 | 94.7% | 23.57 | 59.6 | 0.130 | 2.98 | 159.9 |
| Wasserstein VAR | Adaptation | 680 | 8192 | **0.252** | **100%** | 23.59 | 60.0 | **0.125** | **2.58** | **166.9** |
| MMD VAR | Adaptation | 680 | 8192 | 0.254 | **100%** | **23.68** | **60.4** | **0.125** | 2.68 | 166.2 |

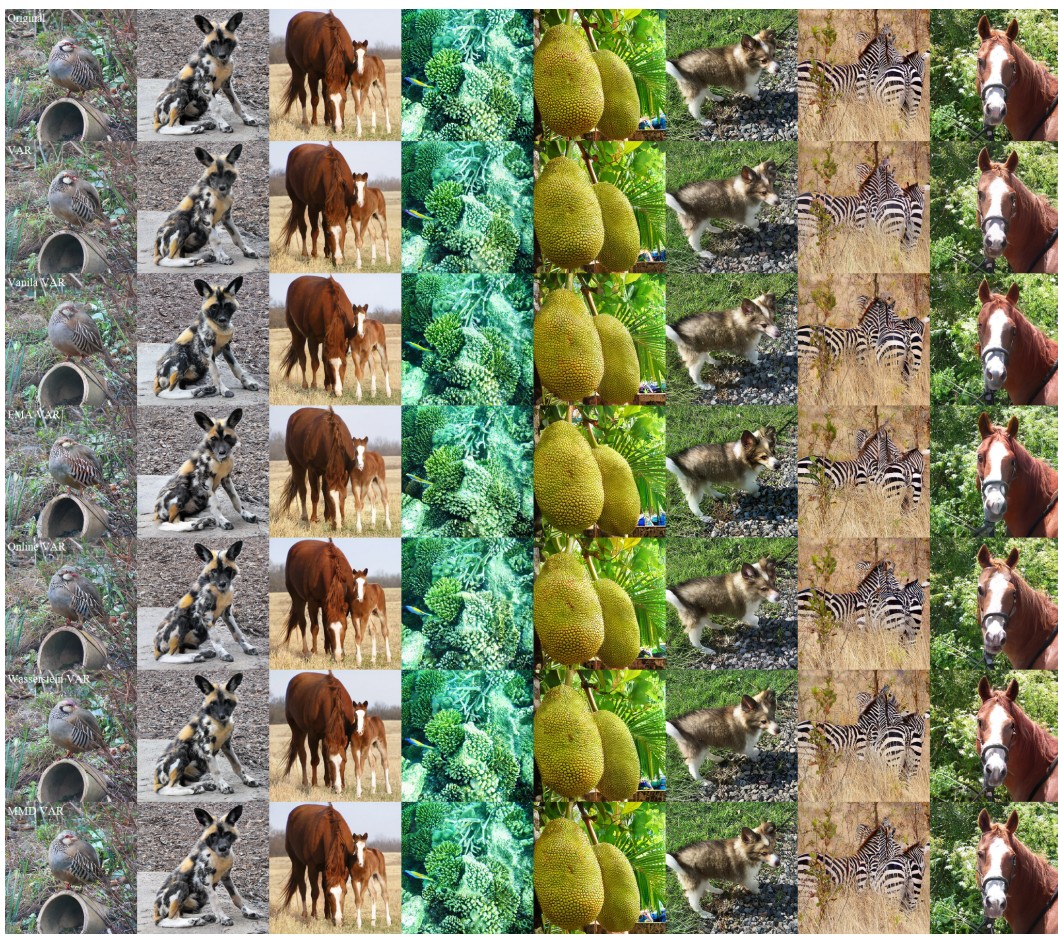

Figure 8: Visualization of reconstructed ImageNet-1k images. Top row: original $256 \times 256$ input images. Subsequent rows (top to bottom): reconstructions from the VAR tokenizer (Tian et al., 2024) and VQ-Transplant-trained Vanilla VAR, EMA VAR, Online VAR, Wasserstein VAR, and MMD VAR models.

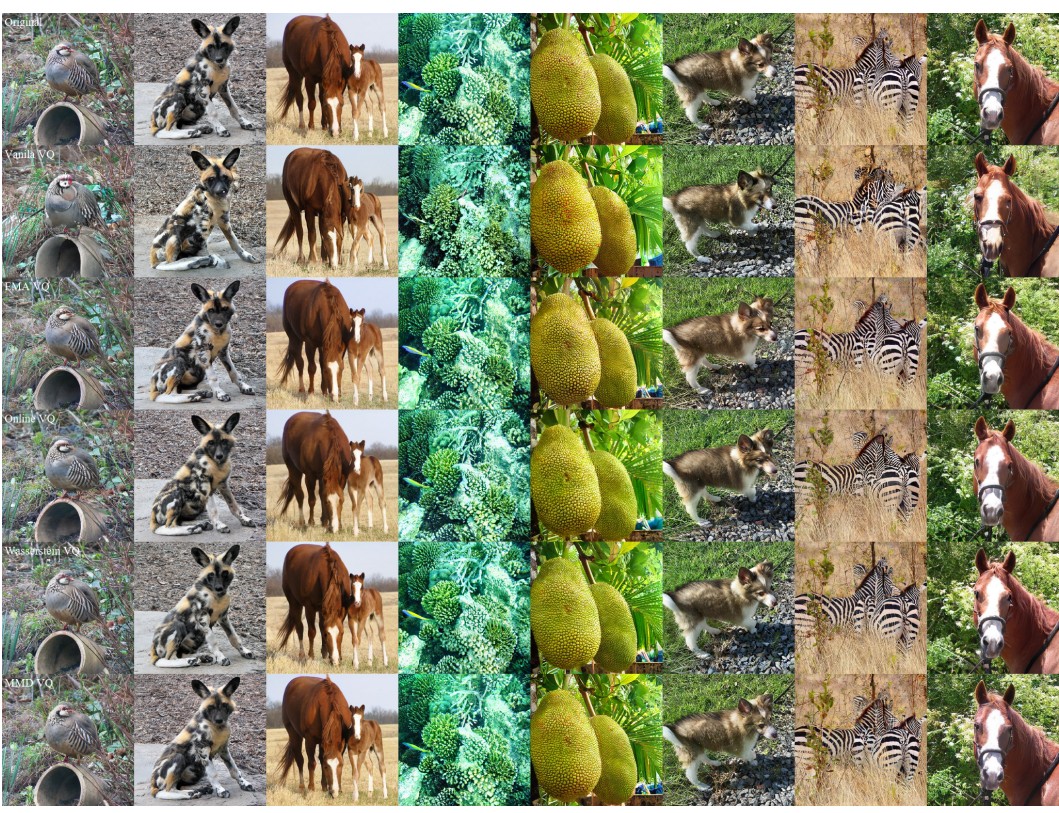

Figure 9: Visualization of reconstructed ImageNet-1k images. Top row: original $256 \times 256$ input images. Subsequent rows (top to bottom): reconstructions from VQ-Transplant-trained Vanilla VQ, EMA VQ, Online VQ, Wasserstein VQ, and MMD VQ models.

