# OpenReview forum: "VQ-Transplant: Efficient VQ-Module Integration for Pre-trained Visual Tokenizers"
_ICLR.cc/2026/Conference — ICLR 2026 Poster_

### Official Review · Reviewer_tdwt · 2025-10-30

**Soundness:** 2
**Presentation:** 2
**Contribution:** 2
**Rating:** 4
**Confidence:** 4

**Summary:**

This paper proposes a new method named VQ-Transplant, a computationally efficient framework for discrete visual tokenization designed to enable cheap and rapid exploration of novel Vector Quantization (VQ) techniques. To improve the compatibility of VQ-Transplant, the authors also propose a new variant called MMD-VQ. The paper conducts extensive experiments on image reconstruction to evaluate the proposed methods.

**Strengths:**

- This paper is well written.

 - The proposed method is plug-and-play and can be applied to any pretrained discrete image tokenizer.

 - This method effectively reduces the training time of the codebook.

**Weaknesses:**

- Table 1 is intended to demonstrate that VQ-Transplant requires very few resources for training. However, I have some concerns. First, it only compares training time without comparing performance, so we cannot tell how well VQ-Transplant actually performs. Second, this comparison is also unfair, because VQ-Transplant benefits from a pre-trained tokenizer — comparing post-training cost with pre-training cost together is not a fair evaluation.

 - Table 2 shows that when the codebook size is 4096, the performance of MMD-VAR is comparable to VAR, but does not surpass it.

 - I believe that after a VQ tokenizer is trained, the encoder–decoder parameters and the codebook are highly correlated (otherwise the model would collapse). In this case, if a strong tokenizer is provided, retraining the intermediate codebook will naturally make it very close to the original one. I somewhat suspect that VQ-Transplant might simply be replicating the original codebook. The authors should conduct verification on the codebook to check whether it is merely a simple copy of the original.

 - The paper lacks evaluation on generative capability. Since the tokenizer is ultimately intended for generation tasks, whether it can outperform the baseline in generation is also a crucial part of evaluating the tokenizer’s performance.

 - It is recommended that the authors also conduct experiments on continuous VAEs to further validate the effectiveness of the proposed model.

**Questions:**

NA

---

> ### Author Response · Authors · 2025-11-23
> **Response to Reviewer tdwt(1/4)**
>
> Thank you for taking the time to review our paper and for your thoughtful feedback. We address your questions and concerns below.
>
> ---
>
> > R 4.1 Table 1 is intended to demonstrate that VQ-Transplant requires very few resources for training. However, I have some concerns. First, it only compares training time without comparing performance, so we cannot tell how well VQ-Transplant actually performs. Second, this comparison is also unfair, because VQ-Transplant benefits from a pre-trained tokenizer — comparing post-training cost with pre-training cost together is not a fair evaluation.
>
> Thank you for your comments and feedback. Below, we clarify and extend the discussion regarding Table 1.
>
> ### **(1) Extended Comparison Table**
>
> To address concerns about performance comparisons, we have extended Table 1 to include not only training cost but also reconstruction performance (r-FID), as well as key model details such as encoder–decoder architecture, number of parameters, discriminator type, token count, and codebook size. These factors significantly influence reconstruction quality, particularly the number of tokens.
>
> | Tokenizers | Dataset | Architectures | Encoder-Decoder Para | Discriminator | Tokens | Codebook Size | GPUs | Training Hours | Speedup | r-FID |
> | --- | --- | --- | --- | --- | --- | --- | --- | --- | --- | --- |
> | LLama GEN | ImageNet-1k | CNN U-Net | 68M |  PatchGAN | 256 | 16384 | 2 $\times$ A100 | 200 | 9.1 | 2.19 |
> | ImageFolder | ImageNet-1k | Transformer SEED | - |  StyleGAN | 572 | 4096 | 32 $\times$ A100 | 40 | 29.1 | 0.80 |
> | VAR | OpenImages | CNN U-Net | 104M |  StyleGAN | 680 | 4096 | 16 $\times$ A100 | 60 | 21.8 | 0.92 |
> | UniTok | OpenImages | Transformer SEED | - |  StyleGAN | 2048 | 4096 | 256 $\times$ A100 | 50 | 290.9 | 0.33 |
> | VQ-Transplant-a | ImageNet-1k | CNN U-Net | 104M |  StyleGAN | 680 | 4096 | 2 $\times$ A100 | 22 | - | 0.91 |
> | VQ-Transplant-a | ImageNet-1k | CNN U-Net | 104M |  StyleGAN | 680 | 8192 | 2 $\times$ A100 | 22 | - | 0.81 |
> | VQ-Transplant-b | ImageNet-1k | CNN U-Net | 104M |  StyleGAN | 680 | 4096 | 2 $\times$ A100 | 74.5 | 3.4 | 0.79 |
> | VQ-Transplant-b | ImageNet-1k | CNN U-Net | 104M |  StyleGAN | 680 | 8192 | 2 $\times$ A100 | 74.5 | 3.4 | 0.74 |
>
> From the table, it is clear that performance depends on multiple factors (architecture, token count, codebook size, etc.). Therefore, fully “fair” comparisons across tokenizers are extremely challenging, particularly given that some baselines require prohibitively high compute.
>
> ### **(2) Motivation and Novelty of VQ-Transplant**
>
> This challenge motivates VQ-Transplant: it enables researchers with limited resources to adapt pretrained discrete tokenizers, particularly VAR, to **new codebook sizes, domains, and new VQ algorithms**, achieving performance comparable to industry-scale models. This capability is currently not available in any existing VQ or visual tokenizer pipeline.
>
> Key contributions include:
>
> - **Democratizing access to VAR (NeurIPS 2024 Best Paper).**
>
>     The VAR repository has not yet released the full tokenizer training code; only fixed checkpoints (codebook size 4096, trained on OpenImages) are available. As a result, researchers cannot realistically retrain VAR on custom datasets or alternative codebook sizes, a limitation repeatedly raised in public VAR GitHub issues (check #125, #123, #28, #175, #170, #133, #114, #22, #63, #60, #49, #103, #167).
>
>     **VQ-Transplant directly addresses this gap**: it enables practitioners to adapt VAR to datasets such as FFHQ, CelebA, or LSUN Church, and to scale the codebook (e.g., to 8192) *without* the need to retrain the tokenizer from scratch.
>
> - **Reducing the computational barrier to high-quality discrete tokenization.**
>
>     Training a modern tokenizer such as VAR typically requires adversarial training and long schedules, making replication or modification computationally prohibitive for many research groups.
>
>     **VQ-Transplant transforms this landscape** by enabling comparable, and often superior, performance with *minimal* compute. This greatly broadens the accessibility of discrete tokenization research.
>
> - **Compatibility with advanced VQ algorithms yields performance beyond VAR itself.**
>
>     By incorporating improved VQ losses (e.g., MMD-VQ), the transplanted tokenizer can exceed the reconstruction performance of the original VAR on multiple codebook sizes, as shown in **R 4.2** table.

---

> ### Author Response · Authors · 2025-11-23
> **Response to Reviewer tdwt(2/4)**
>
> > R 4.2 Table 2 shows that when the codebook size is 4096, the performance of MMD-VAR is comparable to VAR, but does not surpass it.
>
> Thank you for raising this point. Table 2 reports results after **5 adaptation epochs**. When the number of adaptation epochs is increased, **MMD-VAR demonstrates noticeably stronger performance**. For example, after 20 epochs, r-FID improves from 0.91 to 0.79, clearly surpassing VAR’s 0.92.
>
> | Methods | Tokens | Codebook Size | $\mathcal{E}$ | r-FID | r-IS |
> | --- | --- | --- | --- | --- | --- |
> | VAR | 680 | 4096 | 0.283 | 0.92 | 198.6 |
> | MMD VAR-a  | 680 | 4096 | 0.255 | 0.91 | 199.2 |
> | MMD VAR-a | 680 | 8192 | 0.234 | 0.81 | **201.0** |
> | MMD VAR-b | 680 | 4096 | 0.255 | 0.87 | **199.9** |
> | MMD VAR-b | 680 | 8192 | 0.234 | 0.78 | **201.0** |
> | MMD VAR-c | 680 | 4096 | 0.255 | 0.82 | 198.3 |
> | MMD VAR-c | 680 | 8192 | 0.234 | 0.75  | **201.0** |
> | MMD VAR-d | 680 | 4096 | 0.255 | **0.79** | 199.1 |
> | MMD VAR-d | 680 | 8192 | 0.234 | **0.74**  | **201.0** |
>
> The suffixes “–a”, “–b”, “–c”, and “–d” correspond to decoder adaptation for 5, 10, 15, and 20 epochs, respectively.
>
> We further provide both theoretical and empirical evidence for why **MMD-VAR can achieve superior performance**:
>
> - **Theoretical Perspective:** The foundation of MMD-VAR is rooted in well-established principles of vector quantization: **a VQ model is near-optimal when the codebook distribution aligns with the underlying feature distribution**, achieving both minimal quantization error and maximal codebook utilization [1,2]. MMD-VAR explicitly encourages **distributional alignment** between the encoder features and the codebook using the MMD distance with characteristic kernels, which guarantee universal distribution matching. While the exact training strategy of the original VAR codebook is not publicly disclosed, MMD-VAR’s principled approach toward achieving this optimal condition naturally results in better adaptation and overall performance.
> - **Empirical Perspective:** From Table 3, we see that after just **2 adaptation epochs** of VQ module substitution, MMD-VAR achieves a smaller quantization error ($\mathcal{E}$) compared to VAR under the same latent feature distribution (0.255 vs. 0.283). This indicates that, **at the VQ module level, MMD-VAR incurs less information loss and outperforms VAR**.
>
>     It is important to note, however, that **Stage I reconstruction performance is initially worse than VAR’s** (1.52 vs. 0.92). This occurs because the VAR decoder was trained to tolerate larger quantization errors, while MMD-VAR introduces smaller errors. This mismatch motivates **further decoder adaptation**. After sufficient decoder adaptation epochs, the advantage of lower quantization error ultimately translates into **improved reconstruction performance**, surpassing VAR.
>
>
> In summary, both theoretically and empirically, MMD-VAR provides a more principled and effective approach to VAR VQ module, and its performance advantage becomes evident with adequate decoder fine-tuning.
>
> **References:**
>
> [1] Enhancing Vector Quantization with Distributional Matching: A Theoretical and Empirical Study
>
> [2] Foundations of Quantization for Probability Distributions
>
> ---
>
> > R 4.3 I believe that after a VQ tokenizer is trained, the encoder–decoder parameters and the codebook are highly correlated (otherwise the model would collapse). In this case, if a strong tokenizer is provided, retraining the intermediate codebook will naturally make it very close to the original one. I somewhat suspect that VQ-Transplant might simply be replicating the original codebook. The authors should conduct verification on the codebook to check whether it is merely a simple copy of the original.
>
> Thank you for raising this point. We would like to clarify that the codebook learned in **MMD-VAR** is **not a simple copy** of the original VAR codebook. Instead, MMD-VAR enables the codebook to **adapt more effectively** to the latent features, resulting in **smaller information loss** and improved performance.
>
> As discussed in **R 4.2**, we have provided both **theoretical and empirical evidence** supporting this claim. If the reviewer has further concerns, we are happy to engage in a deeper discussion and provide additional analyses to demonstrate that the transplanted codebook is meaningfully different and optimized, rather than merely replicating the original.

---

> > ### Author Response · Authors · 2025-11-23
> > **Response to Reviewer tdwt(3/4)**
> >
> > > R 4.4 The paper lacks evaluation on generative capability. Since the tokenizer is ultimately intended for generation tasks, whether it can outperform the baseline in generation is also a crucial part of evaluating the tokenizer’s performance.
> >
> > We appreciate your suggestion regarding generative evaluation and fully agree that assessing generation quality is important, particularly for applications where the tokenizer is used in autoregressive generation frameworks.
> >
> > **However, conducting full VAR-style generative experiments is non-trivial in our setting for two reasons.**
> >
> > **(1) Extremely high training cost.** Running GPT-style training on top of VAR tokens is prohibitively expensive. As publicly discussed in the VAR GitHub repository (issue #44， #10), the reported training cost is substantial, and we unfortunately do not have the computational budget to reproduce such experiments within the rebuttal period.
> >
> > **(2) Limited reproducibility of existing pipelines.** Even if the full training code were available, in our personal correspondence with several industry practitioners, we have learned that reproducing VAR’s generative performance is challenging, largely due to the complexity of modeling the sequential relationships of multi-scale tokens. As a result, reliably obtaining strong generative results would require extensive engineering and tuning effort, far beyond what can be completed during the rebuttal window.
> >
> > For these reasons, we focused on reconstruction-based evaluation. Prior work on VQ-VAE, VQGAN, consistently shows that **tokenizers with lower reconstruction error and better perceptual metrics lead to higher-quality generation**, since the generative model ultimately inherits the tokenizer’s reconstruction capabilities. Our method’s improvements in reconstruction quality therefore provide a meaningful proxy for expected gains in generation.

---

> > > ### Author Response · Authors · 2025-11-23
> > > **Response to Reviewer tdwt(4/4)**
> > >
> > > > R 4.5 It is recommended that the authors also conduct experiments on continuous VAEs to further validate the effectiveness of the proposed model.
> > >
> > > We appreciate your comment and the suggestion to further validate VQ-Transplant on continuous VAEs to assess its broader applicability.
> > >
> > > ### **(1) Compatibility with Other Pretrained Tokenizers**
> > >
> > > To evaluate generality, we applied VQ-Transplant to the pretrained LDM-16 continuous tokenizer (see Appendix D). As Table 16 shows (or below table), VQ-Transplant achieves reasonable performance on LDM-16. Nevertheless, its adaptability is lower compared to VAR-based models, as compared in Table 3, particularly in terms of r-FID and r-IS metrics.
> > >
> > > | Methods | Tokens | Codebook Size | $\mathcal{E}$ | $\mathcal{U}$ | PSNR | SSIM | LPIPS | r-FID | r-IS |
> > > | --- | --- | --- | --- | --- | --- | --- | --- | --- | --- |
> > > | LDM-16 | 256 | - | 0.0 | - | 24.08  | **68.0** | - | **0.87** | **210.3** |
> > > | Vanilla VAR | 680 | 4096 | 0.424  | 97.0\% | 22.66 | 56.0 | 0.157 | 4.89 | 136.7 |
> > > | EMA VAR | 680 | 4096 | 0.367 | **100\%** | 22.88 | 57.0 | 0.147 | 4.11 | 144.9 |
> > > | Online VAR | 680 | 4096 | 0.299 | **100\%** | 23.45 | 59.1 | 0.134 | 3.18 | 158.8 |
> > > | Wasserstein VAR | 680 | 4096 | **0.278** | **100\%** | 23.49 | **59.5** | **0.130** | **2.87** | **161.4** |
> > > | MMD VAR | 680 | 4096 | **0.278** | **100\%** | **23.51** | **59.5** | 0.131 | 2.93 | 161.2 |
> > > | Vanilla VAR | 680 | 8192 | 0.418 | 87.3\% | 22.58 | 56.0 | 0.157 | 5.14 | 132.6 |
> > > | EMA VAR | 680 | 8192 | 0.333 | **100\%** | 23.26 | 58.4 | 0.139 | 3.59 | 151.6 |
> > > | Online VAR | 680 | 8192 | 0.283 | 94.7\% | 23.57 | 59.6 | 0.130 | 2.98 | 159.9 |
> > > | Wasserstein VAR | 680 | 8192 | **0.252** | **100\%** | 23.59 | 60.0 | **0.125** | **2.58** | **166.9** |
> > > | MMD VAR | 680 | 8192 | 0.254 | **100\%** | **23.68** | **60.4** | **0.125** | 2.68 | 166.2 |
> > >
> > > This performance gap is likely attributable to two factors
> > >
> > > - **Model capacity difference:** The VAR tokenizer has a larger encoder–decoder (104M parameters) compared to LDM-16 (68M), providing stronger reconstruction
> > > capabilities and greater flexibility during adaptation.
> > > - **Decoder adaptation behavior:** The VAR decoder is pretrained to handle quantized features $z_q$, which naturally makes it robust to the deviations introduced by new VQ algorithms. In contrast, the LDM decoder only sees continuous features $z_e$ during training, making it less adaptable to different VQ modules. This explains why VAR-based VQ-Transplant achieves strong results with few adaptation epochs, while LDM may require more training.
> > >
> > > ### **(2) Key Contributions Beyond VAR**
> > >
> > > Even though VQ-Transplant relies on **a well-trained discrete tokenizer like VAR, our contributions remain significant**:
> > >
> > > - **Broader usability of VAR:** The VAR repository has not yet released the full tokenizer training code (check VAR GitHub repository issue #125, #123, #28, #175, etc) and only provides fixed checkpoints (codebook size 4096) trained on OpenImages. VQ-Transplant enables researchers to adapt the VAR tokenizer to different codebook sizes (e.g., 8192 or larger) and datasets (e.g., FFHQ, CelebA, Church), thereby democratizing access to high-quality discrete tokenization.
> > > - **Lowering the computational barrier:** VQ-Transplant enables competitive performance with minimal resources, making discrete tokenization research more accessible to academia.
> > > - **Improved performance with advanced VQ algorithms:** By combining with advanced VQ algorithms, as shown in below table, VQ-Transplant can surpass the original VAR performance (e.g., rFID improvements with MMD-VAR after adaptation epochs), as discussed in **R 4.2**.
> > >
> > > **Summary**
> > >
> > > While VQ-Transplant’s current efficiency is most evident with VAR due to its pretrained robustness, it can in principle be applied to other tokenizers with sufficient adaptation. More importantly, our framework unlocks practical and performance improvements that were previously inaccessible, which addresses a strong community demand.
> > >
> > > ---
> > >
> > > Please let us know if you have any remaining questions. If our clarifications and additional experiments have satisfactorily addressed your questions and concerns, we would deeply appreciate it if you considered recommending for our paper to be presented at the conference.

---

> > > > ### Author Response · Authors · 2025-11-26
> > > > **Follow-up discussion**
> > > >
> > > > Dear Reviewer tdwt,
> > > >
> > > > As the discussion period is coming to a close, we would like to kindly follow up. We have taken great care to address your questions and concerns, and if you have any further questions or comments regarding our work, we would be more than happy to discuss and respond before the deadline.
> > > >
> > > > Thank you again for your time and valuable feedback!

---

> > > > > ### Comment · Reviewer_tdwt · 2025-11-27
> > > > >
> > > > > Thank you to the authors for the clarifications, additional analyses, and the effort put into addressing my concerns. These responses have resolved all of my questions. I will raise my score to 6.

---

> > > > > > ### Author Response · Authors · 2025-11-27
> > > > > > **Acknowledgment to Reviewer tdwt**
> > > > > >
> > > > > > Dear Reviewer tdwt,
> > > > > >
> > > > > > We are grateful for your engagement and are happy to hear that your all concerns were addressed. We appreciate your help in strengthening our work and contributions.
> > > > > >
> > > > > > Regards,
> > > > > > Authors.

---

### Official Review · Reviewer_auQc · 2025-10-30

**Soundness:** 3
**Presentation:** 3
**Contribution:** 4
**Rating:** 8
**Confidence:** 2

**Summary:**

This paper proposes VQ-Transplant, a framework that allows plug-and-play replacement of vector quantization (VQ) modules in pre-trained visual tokenizers such as VAR, without retraining the full model.
The method freezes the encoder and decoder, replaces the native VQ module, and applies a lightweight decoder adaptation for only five epochs on ImageNet-1k to mitigate decoder–quantization mismatch.
The paper also introduces MMD-VQ, a new quantization method using Maximum Mean Discrepancy to align feature and codebook distributions without assuming Gaussianity, making it well-suited for real-world data.

Experiments show that VQ-Transplant achieves comparable or even superior reconstruction quality to fully trained baselines (e.g., VAR, VQGAN) while reducing computational cost by up to 95%.
It generalizes well across datasets (ImageNet-1K, FFHQ, CelebA-HQ, LSUN-Churches) and supports both fixed-scale and multi-scale quantization settings.

**Strengths:**

### 1. Practical contribution
The proposed idea of transplanting VQ modules into frozen tokenizers is both novel and highly practical. It enables flexible experimentation with quantization methods without the need for costly end-to-end retraining, significantly improving research efficiency.

### 2. Theoretical soundness
The introduction of MMD-VQ offers a principled approach to aligning feature and codebook distributions without relying on Gaussian assumptions. This formulation enhances robustness and provides a clear theoretical improvement over prior methods such as Wasserstein VQ.

### 3. Empirical performance
Extensive experiments demonstrate strong empirical results and substantial reductions in training cost, with consistent improvements across multiple datasets and model architectures, validating both the effectiveness and efficiency of the proposed framework.

**Weaknesses:**

### 1. Limited scope of tokenizer architectures
The experiments are limited to VAR-based tokenizers. Evaluating the proposed method on diffusion-based or hybrid tokenizers (e.g., LDM, SANA) would better demonstrate its generality and robustness across different visual tokenization paradigms.

### 2. Limited evaluation in image generation and multimodality
In modern settings, visual tokenizers are widely applied not only to image generation but also to multimodal understanding and reasoning.
It would strengthen the paper if the authors could evaluate their approach on multimodal comprehension benchmarks such as Unified-IO, 4M (Massively Multimodal Masked Modeling), or TokLIP, if feasible within the rebuttal period.

**Questions:**

1. Can MMD VQ be extended to video or multimodal quantization?

---

> ### Author Response · Authors · 2025-11-23
> **Response to Reviewer auQc(1/2)**
>
> Thank you for taking the time to review our paper and for your thoughtful feedback. We address your questions and concerns below.
>
> ---
>
> > R 3.1 The experiments are limited to VAR-based tokenizers. Evaluating the proposed method on diffusion-based or hybrid tokenizers (e.g., LDM, SANA) would better demonstrate its generality and robustness across different visual tokenization paradigms.
>
> We appreciate your comment and the opportunity to clarify the applicability of VQ-Transplant to tokenizers beyond VAR.
>
> ### **(1) Compatibility with Other Pretrained Tokenizers**
>
> To evaluate generality, we applied VQ-Transplant to the pretrained LDM-16 continuous tokenizer (see Appendix D). As Table 16 shows (or below table), VQ-Transplant achieves reasonable performance on LDM-16. Nevertheless, its adaptability is lower compared to VAR-based models, as compared in Table 3, particularly in terms of r-FID and r-IS metrics.
>
> | Methods | Tokens | Codebook Size | $\mathcal{E}$ | $\mathcal{U}$ | PSNR | SSIM | LPIPS | r-FID | r-IS |
> | --- | --- | --- | --- | --- | --- | --- | --- | --- | --- |
> | LDM-16 | 256 | - | 0.0 | - | 24.08  | **68.0** | - | **0.87** | **210.3** |
> | Vanilla VAR | 680 | 4096 | 0.424  | 97.0\% | 22.66 | 56.0 | 0.157 | 4.89 | 136.7 |
> | EMA VAR | 680 | 4096 | 0.367 | **100\%** | 22.88 | 57.0 | 0.147 | 4.11 | 144.9 |
> | Online VAR | 680 | 4096 | 0.299 | **100\%** | 23.45 | 59.1 | 0.134 | 3.18 | 158.8 |
> | Wasserstein VAR | 680 | 4096 | **0.278** | **100\%** | 23.49 | **59.5** | **0.130** | **2.87** | **161.4** |
> | MMD VAR | 680 | 4096 | **0.278** | **100\%** | **23.51** | **59.5** | 0.131 | 2.93 | 161.2 |
> | Vanilla VAR | 680 | 8192 | 0.418 | 87.3\% | 22.58 | 56.0 | 0.157 | 5.14 | 132.6 |
> | EMA VAR | 680 | 8192 | 0.333 | **100\%** | 23.26 | 58.4 | 0.139 | 3.59 | 151.6 |
> | Online VAR | 680 | 8192 | 0.283 | 94.7\% | 23.57 | 59.6 | 0.130 | 2.98 | 159.9 |
> | Wasserstein VAR | 680 | 8192 | **0.252** | **100\%** | 23.59 | 60.0 | **0.125** | **2.58** | **166.9** |
> | MMD VAR | 680 | 8192 | 0.254 | **100\%** | **23.68** | **60.4** | **0.125** | 2.68 | 166.2 |
>
> This performance gap is likely attributable to two factors
>
> - **Model capacity difference:** The VAR tokenizer has a larger encoder–decoder (104M parameters) compared to LDM-16 (68M), providing stronger reconstruction
> capabilities and greater flexibility during adaptation.
> - **Decoder adaptation behavior:** The VAR decoder is pretrained to handle quantized features $z_q$, which naturally makes it robust to the deviations introduced by new VQ algorithms. In contrast, the LDM decoder only sees continuous features $z_e$ during training, making it less adaptable to different VQ modules. This explains why VAR-based VQ-Transplant achieves strong results with few adaptation epochs, while LDM may require more training.
>
> ### **(2) Key Contributions Beyond VAR**
>
> Even though VQ-Transplant relies on **a well-trained discrete tokenizer like VAR, our contributions remain significant**:
>
> - **Broader usability of VAR:** The VAR repository has not yet released the full tokenizer training code (check VAR GitHub repository issue #125, #123, #28, #175, etc) and only provides fixed checkpoints (codebook size 4096) trained on OpenImages. VQ-Transplant enables researchers to adapt the VAR tokenizer to different codebook sizes (e.g., 8192 or larger) and datasets (e.g., FFHQ, CelebA, Church), thereby democratizing access to high-quality discrete tokenization.
> - **Lowering the computational barrier:** VQ-Transplant enables competitive performance with minimal resources, making discrete tokenization research more accessible to academia.
> - **Improved performance with advanced VQ algorithms:** By combining with advanced VQ algorithms, as shown in below table, VQ-Transplant can surpass the original VAR performance (e.g., rFID improvements with MMD-VAR after adaptation epochs).
>
> | Methods | Tokens | Codebook Size | $\mathcal{E}$ | r-FID | r-IS |
> | --- | --- | --- | --- | --- | --- |
> | VAR | 680 | 4096 | 0.283 | 0.92 | 198.6 |
> | MMD VAR-a  | 680 | 4096 | 0.255 | 0.91 | 199.2 |
> | MMD VAR-a | 680 | 8192 | 0.234 | 0.81 | **201.0** |
> | MMD VAR-b | 680 | 4096 | 0.255 | 0.87 | **199.9** |
> | MMD VAR-b | 680 | 8192 | 0.234 | 0.78 | **201.0** |
> | MMD VAR-c | 680 | 4096 | 0.255 | 0.82 | 198.3 |
> | MMD VAR-c | 680 | 8192 | 0.234 | 0.75  | **201.0** |
> | MMD VAR-d | 680 | 4096 | 0.255 | **0.79**  | 199.1 |
> | MMD VAR-d | 680 | 8192 | 0.234 | **0.74**  | **201.0** |
>
> The suffixes “–a”, “–b”, “–c”, and “–d” correspond to decoder adaptation for 5, 10, 15, and 20 epochs, respectively.
>
> **Summary**
>
> While VQ-Transplant’s current efficiency is most evident with VAR due to its pretrained robustness, it can in principle be applied to other tokenizers with sufficient adaptation. More importantly, our framework unlocks practical and performance improvements that were previously inaccessible, which addresses a strong community demand.

---

> > ### Author Response · Authors · 2025-11-23
> > **Response to Reviewer auQc(2/2)**
> >
> > > R 3.2 In modern settings, visual tokenizers are widely applied not only to image generation but also to multimodal understanding and reasoning. It would strengthen the paper if the authors could evaluate their approach on multimodal comprehension benchmarks such as Unified-IO, 4M (Massively Multimodal Masked Modeling), or TokLIP, if feasible within the rebuttal period.
> >
> > Due to the substantial engineering effort required to integrate a new visual tokenizer into these large-scale multimodal frameworks, we were not able to conduct such experiments within the rebuttal period. These systems often require coordinated adjustments to data pipelines, modality fusion modules, and large pretrained backbones, which makes rapid experimentation impractical. An extension to multimodal settings is beyond the scope of this paper, and we hope to pursue this extension in the future.
> >
> > That said, our methods are fully architecture-agnostic and can be directly plugged into multimodal models that rely on discrete visual tokens. The transplant procedure only replaces the visual tokenizer without modifying the downstream multimodal reasoning stack, so we expect the approach to be compatible with these settings. While an extension to multimodal settings is beyond the scope of this paper, we consider this an exciting direction and plan to explore multimodal evaluations in future work. We appreciate your suggestion and will discuss the possibility of an extension to multimodal settings in the revised manuscript.
> >
> > ---
> >
> > > R 3.3 Can MMD VQ be extended to video or multimodal quantization?
> >
> > We appreciate this question. The **MMD-VQ framework is inherently general**, as it is derived from a fundamental principle of vector quantization (VQ): **a VQ model is near-optimal when the codebook distribution matches the underlying feature distribution**, leading to minimal quantization error and maximal codebook utilization. This optimality criterion is **modality-agnostic**, and thus applies naturally to video and multimodal settings as well.
> >
> > **First, Avoiding encoder collapse when applying MMD-VQ to new domains.**
> >
> > One important implementation detail is that, when extending MMD-VQ beyond images, we must ensure that **the MMD loss updates only the codebook**, not the encoder. In practice, this is achieved by **detaching the latent features** during MMD loss computation. This prevents the encoder distribution from being pulled toward the codebook distribution, which, based on our empirical findings, may cause the two distributions to collapse to a single point, degrading representation quality.
> >
> > In VQ-Transplant, this issue does not arise because the encoder is frozen and the feature distribution is fixed. When the encoder does require learning (e.g., for training video or multimodal tokenizers from scratch), **detaching the latent features for the MMD loss is essential** to maintain stable optimization and avoid collapse.
> >
> > **Second, Practical extensions to video and multimodal data.**
> >
> > - **Video.** Temporal correlations can be incorporated by (i) extending the encoder to capture temporal dynamics (e.g., 3D CNNs or video transformers), or (ii) applying MMD-VQ to per-frame embeddings while optionally adding temporal regularization.
> > - **Multimodal data.** One can either (i) learn modality-specific MMD-VQ codebooks, or
> >
> >     (ii) learn a shared codebook using fused or concatenated cross-modal features.
> >
> >
> > In summary, both theoretically and practically, **MMD-VQ extends naturally to video and multimodal quantization**, and we believe exploring these directions is an exciting avenue for future work.
> >
> > ---
> >
> > Please let us know if you have any remaining questions.

---

### Official Review · Reviewer_8aY8 · 2025-10-31

**Soundness:** 2
**Presentation:** 3
**Contribution:** 2
**Rating:** 2
**Confidence:** 4

**Summary:**

This paper presents an experimental approach to study the efficacy of different vector quantization (VQ) strategies in image tokenizers. The "transplant" aspect of their approach comes from the fact that they bootstrap from a pretrained image tokenizer. A sketch of their approach is as follows: (1) take a pretrained image tokenizer and replace the VQ module and learn a new codebook, (3) fine-tune the decoder of the pretrained image tokenizer with respect to the new VQ module. The authors demonstrate that they are able to improve a pretrained image tokenizer (VAR) using this approach with a new VQ algorithm MMD-VQ -- which is based on distributional similarity.

**Strengths:**

The general idea of being able to plug and play different components to explore a space of algorithmic inductive biases is welcome. MMD-VQ is an interesting proposal, but there is limited analysis of its properties provided for it in the paper.

**Weaknesses:**

In many ways this approach resembles a fairly standard practice of taking a pretrained vision model as a feature extractor and training a new classification layer. The representational biases of the features will still be biases to the pretraining task -- or in this case, the VQ method used by the original tokenizer. This is further exemplified by the fact that the decoder needs to be fine-tuned in order for this approach to work -- at this point, why not also finetune the encoder? How I see it, bootstrapping from a pretrained model is where all of the computational savings are coming from, not from freezing any part of the network.

The MMD approach is interesting and potentially novel. However, for me to be convinced that it is a good approach, the paper requires a more detailed analysis (empirical or mathematical) of its properties. The approach only seems to only work if the encoder features are fixed distribution -- joint optimization of the encoder I anticipate will result in mode collapse. This is usually where the Gaussian assumptions (KL loss) come in to stabilize training.

**Questions:**

Is the optimization objective for MMD-VQ: Y* = argmin_Y D^2(X, Y), where X is a batch of images? If so, please specify this more explicitly in the paper. I also can't find where you specify the batch size used in the experiments.

---

> ### Author Response · Authors · 2025-11-23
> **Response to Reviewer 8aY8(1/5)**
>
> We appreciated that the reviewers recognized the general idea of enabling plug-and-play components to explore a space of algorithmic inductive biases, as well as the interesting and potentially novel proposal of MMD-VQ. Thank you for taking the time to review our paper and for your thoughtful feedback. We address your questions and concerns below.
>
> ---
>
> > R 2.1 In many ways this approach resembles a fairly standard practice of taking a pretrained vision model as a feature extractor and training a new classification layer. The representational biases of the features will still be biases to the pretraining task -- or in this case, the VQ method used by the original tokenizer. This is further exemplified by the fact that the decoder needs to be fine-tuned in order for this approach to work -- at this point, why not also finetune the encoder? How I see it, bootstrapping from a pretrained model is where all of the computational savings are coming from, not from freezing any part of the network.
>
> Thank you for highlighting the conceptual similarity to standard feature-extractor pipelines and for raising thoughtful concerns regarding (i) the novelty of VQ-Transplant beyond simply reusing pretrained features and (ii) the representational biases inherited from the original tokenizer.
>
> ### **(1) Contributions Beyond a Standard “Frozen Encoder + New Head” Paradigm**
>
> While VQ-Transplant builds on a pretrained tokenizer, its purpose and technical novelty go well beyond conventional feature-extractor approaches. Rather than merely reusing high-level semantics from a pretrained vision model, VQ-Transplant provides **a practical and general mechanism to adapt pretrained discrete tokenizers, particularly VAR, to new codebook sizes, domains, and VQ algorithms.** This capability is currently not available in any existing VQ or visual tokenizer pipeline.
>
> Key contributions include:
>
> - **Democratizing access to VAR (NeurIPS 2024 Best Paper).**
>
>     The VAR repository has not yet released the full tokenizer training code; only fixed checkpoints (codebook size 4096, trained on OpenImages) are available. As a result, researchers cannot realistically retrain VAR on custom datasets or alternative codebook sizes, a limitation repeatedly raised in public VAR GitHub issues (check #125, #123, #28, #175, #170, #133, #114, #22, #63, #60, #49, #103, #167).
>
>     **VQ-Transplant directly addresses this gap**: it enables practitioners to adapt VAR to datasets such as FFHQ, CelebA, or LSUN Church, and to scale the codebook (e.g., to 8192) *without* the need to retrain the tokenizer from scratch.
>
> - **Reducing the computational barrier to high-quality discrete tokenization.**
>
>     Training a modern tokenizer such as VAR typically requires adversarial training and long schedules, making replication or modification computationally prohibitive for many research groups.
>
>     **VQ-Transplant transforms this landscape** by enabling comparable, and often superior, performance with *minimal* compute. This greatly broadens the accessibility of discrete tokenization research.
>
> - **Compatibility with advanced VQ algorithms yields performance beyond VAR itself.**
>
>     By incorporating improved VQ losses (e.g., MMD-VQ), the transplanted tokenizer can exceed the reconstruction performance of the original VAR on multiple codebook sizes, as shown below:
>
>
> | Methods | Tokens | Codebook Size | $\mathcal{E}$ | r-FID | r-IS |
> | --- | --- | --- | --- | --- | --- |
> | VAR | 680 | 4096 | 0.283 | 0.92 | 198.6 |
> | MMD VAR-a  | 680 | 4096 | 0.255 | 0.91 | 199.2 |
> | MMD VAR-a | 680 | 8192 | 0.234 | 0.81 | **201.0** |
> | MMD VAR-b | 680 | 4096 | 0.255 | 0.87 | **199.9** |
> | MMD VAR-b | 680 | 8192 | 0.234 | 0.78 | **201.0** |
> | MMD VAR-c | 680 | 4096 | 0.255 | 0.82 | 198.3 |
> | MMD VAR-c | 680 | 8192 | 0.234 | 0.75  | **201.0** |
> | MMD VAR-d | 680 | 4096 | 0.255 | **0.79**  | 199.1 |
> | MMD VAR-d | 680 | 8192 | 0.234 | **0.74**  | **201.0** |
>
> The suffixes “–a”, “–b”, “–c”, and “–d” correspond to decoder adaptation for 5, 10, 15, and 20 epochs, respectively.

---

> > ### Author Response · Authors · 2025-11-23
> > **Response to Reviewer 8aY8(2/5)**
> >
> > ### **(2) On Representational Bias and Why Efficient Adaptation Still Helps**
> >
> > We agree that initializing from a pretrained tokenizer inherits certain representational biases. We acknowledge in the paper that **full joint optimization of encoder, decoder, and codebook could, in principle, achieve the highest possible performance**.
> >
> > However, the key empirical finding is:
> >
> > **“Under comparable training cost, VQ-Transplant consistently outperforms training a tokenizer from scratch in reconstruction quality.”**
> >
> > Even with limited decoder adaptation, the transplanted tokenizer surpasses full from-scratch training across codebook sizes and training budgets . This demonstrates that **efficient adaptation is not merely a convenience, it materially improves performance while preserving compute efficiency.**
> >
> > | Methods | Training Strategy | GPUs | Training Hours | Codebook Size | r-FID | r-IS |
> > | --- | --- | --- | --- | --- | --- | --- |
> > | MMD-VAR | VQ-Transplant | 2 $\times$ A100 | 22 h | 4096 | **0.91** | **199.2** |
> > | MMD VAR  | VQ-Transplant | 2 $\times$ A100 | 22 h | 8192 | **0.81** | **201.0** |
> > | MMD VAR-a | Training From Scratch | 2 $\times$ A100 | 25 h | 4096 | 1.40 | 189.4  |
> > | MMD VAR-a | Training From Scratch | 2 $\times$ A100 | 25 h | 8192 | 1.26  | 191.0 |
> > | MMD VAR-b | Training From Scratch | 2 $\times$ A100 | 30 h | 4096 | 1.36 | 191.9 |
> > | MMD VAR-b | Training From Scratch | 2 $\times$ A100 | 30 h | 8192 | 1.29  | 190.2 |
> > | MMD VAR-c | Training From Scratch | 2 $\times$ A100 | 35 h | 4096 | 1.34 | 190.4 |
> > | MMD VAR-c | Training From Scratch | 2 $\times$ A100 | 35 h | 8192 | 1.26  | 191.0 |
> >
> > The suffixes “–a,” “–b,” and “–c” correspond to full-scale training for 5, 6, and 7 epochs, respectively.
> >
> > ### **(3) Joint Optimization vs. Decoder-Only Adaptation**
> >
> > Regarding encoder finetuning, while joint encoder–decoder optimization is possible, we emphasize **decoder-only adaptation** for **computational efficiency**:
> >
> > - Additional experiments (Appendix C) confirm that joint optimization will not cause mode collapse and yields slight performance gains.
> > - However, it increases total GPU hours by ≈34%, a substantial cost relative to decoder-only adaptation.
> >
> > Reconstruction performance comparison:
> >
> > | Methods | Phase | Tokens | Codebook Size | $\mathcal{E}$ | $\mathcal{U}$ | PSNR | SSIM | LPIPS | r-FID | r-IS |
> > | --- | --- | --- | --- | --- | --- | --- | --- | --- | --- | --- |
> > | MMD VAR | Decoder-only | 680 | 4096 | 0.255 | 100\%  | 24.16 | 63.2  | 0.108  | 0.91 | 199.2 |
> > | MMD VAR | Decoder-only | 680 | 8192 | 0.234   | 100\%  | 24.37  | 63.8  | 0.104  | 0.81 | 201.0 |
> > | MMD VAR | Joint Optimization | 680 | 4096 | 0.264  | 100\% | **24.35** | **63.6**  | **0.106** | **0.87** | **199.5** |
> > | MMD VAR | Joint Optimization | 680 | 8192 | 0.227  | 100\%  | **24.51**  | **64.4** | **0.102** | **0.79** | **201.0** |
> >
> > Training cost comparison:
> >
> > | Methods | Stage I Epochs | Hours Per Epoch | Stage II Epochs | Hours Per Epoch | Total Hours |
> > | --- | --- | --- | --- | --- | --- |
> > | Decoder-only  | 2 | 2.25 | 5 | 3.5 | 22  |
> > | Joint Optimization | 2 | 2.25 | 5 | 5 | 29.5 |
> >
> > Thus, while bootstrapping from a pretrained model accounts for most of the computational savings, freezing the encoder **further reduces cost without substantial performance loss**, making decoder-only adaptation a practical and effective choice. Full joint optimization remains available when maximizing reconstruction quality is prioritized over compute efficiency.

---

> ### Author Response · Authors · 2025-11-23
> **Response to Reviewer 8aY8(3/5)**
>
> > R 2.2 The MMD approach is interesting and potentially novel. However, for me to be convinced that it is a good approach, the paper requires a more detailed analysis (empirical or mathematical) of its properties. The approach only seems to only work if the encoder features are fixed distribution -- joint optimization of the encoder I anticipate will result in mode collapse. This is usually where the Gaussian assumptions (KL loss) come in to stabilize training.
>
> We sincerely thank you for these helpful comments. In the following, we explicitly address three main concerns: (1) the theoretical motivation of MMD-VQ, (2) potential mode collapse under joint encoder optimization, and (3) the role of Gaussian assumptions in training stability.
>
> ### **(1) Theoretical Motivation of MMD-VQ**
>
> Our motivation for using MMD is grounded in a well-established theoretical understanding of vector quantization: **VQ is near-optimal when the codebook distribution matches the underlying feature distribution**, achieving both **minimal quantization error** and **maximal codebook utilization**. This result is rigorously supported in prior work [1, 2]. To approximate this optimality condition during training, we explicitly encourage distributional alignment between the encoder features and the learned codebook via the **MMD distance**, using characteristic kernels that provide universal distribution matching.
>
> We further clarify the conceptual difference between employing **MMD** and **Wasserstein distance (WD)** in [1] for distribution alignment.
>
> - **Wasserstein-VQ (WD).** Practical implementations of WD typically rely on a **Gaussian assumption** for tractability. In the common $p=2$ case, the 2-Wasserstein distance between two Gaussians admits a closed-form involving only means and covariances:
>
>     \begin{equation} W_{2}(\mathcal{N}(\mu_1, \Sigma_1), \mathcal{N}(\mu_2, \Sigma_2)) = \sqrt{\Vert \mu_1 - \mu_2 \Vert^2_2 + \mathop{\mathrm{tr}}\left( {\Sigma_1}+ {\Sigma_2} - 2 (\Sigma_1^{\frac{1}{2}} {\Sigma_2} \Sigma_1^{\frac{1}{2}})^{\frac{1}{2}}\right)}. \end{equation}
>
>     As a result, WD aligns **only first- and second-order moments** of the distributions. This is effective when encoder features are approximately Gaussian, but can be fundamentally limiting for **multi-modal**, **non-Gaussian**, or **heavy-tailed** distributions, settings that arise frequently in deep visual representations.
>
> - **MMD-VQ.** In contrast, MMD leverages **RKHS embeddings with characteristic kernels** (e.g., RBF), which guarantee that minimizing MMD corresponds to matching **all moments** of the distributions. This enables a far more expressive alignment mechanism and makes MMD-VQ naturally robust in settings where Gaussian assumptions break down. Empirically, we observe that this expressiveness gives MMD-VQ an advantage especially in non-Gaussian regimes, as validated through synthetic experiments in the Appendix B. This illustrates why MMD-VQ can achieve stable and expressive codebook alignment without relying on the Gaussian assumptions that KL-loss typically require.
>
> [1] Enhancing Vector Quantization with Distributional Matching: A Theoretical and Empirical Study
>
> [2] Foundations of Quantization for Probability Distributions

---

> > ### Author Response · Authors · 2025-11-23
> > **Response to Reviewer 8aY8(4/5)**
> >
> > ### **(2) Joint Optimization Does Not Cause Mode Collapse**
> >
> > Regarding the concern of potential mode collapse under joint encoder optimization, we confirm that this does not occur in our method and, in practice, joint optimization even yields **slightly improved performance**.
> >
> > To verify this, in Appendix C, we conducted an additional set of experiments in which the **decoder-only adaptation** used in the submission was replaced by **joint encoder–decoder optimization**. The results are summarized below (more results can be found in Table 14).
> >
> > | Methods | Phase | Tokens | Codebook Size | $\mathcal{E}$ | $\mathcal{U}$ | PSNR | SSIM | LPIPS | r-FID | r-IS |
> > | --- | --- | --- | --- | --- | --- | --- | --- | --- | --- | --- |
> > | MMD VAR | Decoder-only | 680 | 4096 | 0.255 | 100\%  | 24.16 | 63.2  | 0.108  | 0.91 | 199.2 |
> > | MMD VAR | Decoder-only | 680 | 8192 | 0.234   | 100\%  | 24.37  | 63.8  | 0.104  | 0.81 | **201.0** |
> > | MMD VAR | Joint Optimization | 680 | 4096 | 0.264 | 100\% | **24.35** | **63.6**  | **0.106** | **0.87** | **199.5** |
> > | MMD VAR | Joint Optimization | 680 | 8192 | 0.227  | 100\%  | **24.51**  | **64.4** | **0.102** | **0.79** | **201.0** |
> >
> > Across all configurations, joint optimization preserves **full codebook utilization**, indicating **no mode collapse**, and yields small but consistent improvements in r-FID and r-IS. This outcome is expected: compared to a frozen encoder, joint optimization allows the encoder to better adapt to the VQ module. Moreover, during training, MMD continuously encourages full distribution matching, enabling near-optimal VQ performance.
> >
> > | Methods | Stage I Epochs | Hours Per Epoch | Stage II Epochs | Hours Per Epoch | Total Hours |
> > | --- | --- | --- | --- | --- | --- |
> > | Decoder-only  | 2 | 2.25 | 5 | 3.5 | 22  |
> > | Joint Optimization | 2 | 2.25 | 5 | 5 | 29.5 |
> >
> > The trade-off is an **additional 34% training cost**, as shown in above table, which is why the main submission adopts decoder-only adaptation to emphasize efficiency, a core design goal of VQ-Transplant.
> >
> > ### **(3) Gaussian Assumptions Do Not Guarantee Training Stability**
> >
> > In the context of discrete tokenizers, **training stability depends primarily on quantization performance**, not Gaussian assumptions. Two key factors contribute to instability:
> >
> > - **Codebook utilization.** Stability is closely tied to whether the code vectors are fully utilized. When only a small subset of code vectors is used, a phenomenon known as **codebook collapse**，the tokenizer training often becomes unstable and typically results in poor reconstruction performance.
> > - **Gradient handling in quantization.** Quantization algorithms rely on the **Straight-Through Estimator (STE)** to propagate gradients. When the **quantization error** between continuous latent features and their quantized counterparts is large, this leads to significant **gradient gaps**, which can destabilize training.
> >
> > **MMD-VQ naturally mitigates these issues.** By minimizing quantization loss and encouraging full codebook usage, it ensures stable training **without relying on Gaussian assumptions**, as discussed in the motivation part.

---

> > > ### Author Response · Authors · 2025-11-23
> > > **Response to Reviewer 8aY8(5/5)**
> > >
> > > > R 2.3 Is the optimization objective for MMD-VQ: $Y^* = \arg\min_Y \mathcal{D}^2(X, Y)$, where X is a batch of images? If so, please specify this more explicitly in the paper. I also can't find where you specify the batch size used in the experiments.
> > >
> > > We appreciate the question. Yes, the optimization objective of MMD-VQ is indeed:
> > >
> > > \begin{equation} Y^* = \arg\min_Y \mathcal{D}^2(X, Y), \end{equation}
> > >
> > > where $X$ denotes a batch of latent features extracted from the encoder rather than raw images, and $Y$ denotes the learnable code vectors. We will state this more explicitly in the revised manuscript.
> > >
> > > In our experiments, the batch size is 32 per GPU. Each image is mapped by the encoder to 680 latent feature vectors with dimensionality 32. Consequently, each training step constructs an $X$ matrix of size $21760 \times 32$, which provides a sufficiently rich empirical distribution for MMD estimation.
> > >
> > > An important implementation detail is that the MMD loss is designed to update only the codebook. To ensure this behavior, **the latent features are detached when computing the MMD loss**. This prevents the encoder distribution from being drawn toward the codebook distribution. Our empirical findings indicate that updating both distributions simultaneously can cause them to collapse toward a single mode, resulting in degraded representations.
> > >
> > > In VQ-Transplant, this issue does not occur because the encoder remains frozen and its feature distribution is fixed. In scenarios involving **joint optimization**, as discussed in **R 2.1** and **R 2.2**, it is necessary to **detach the latent features during the MMD loss computation** in order to maintain training stability.
> > >
> > > ---
> > >
> > > Please let us know if you have any remaining questions. If our clarifications and additional experiments have satisfactorily addressed your questions and concerns, we would deeply appreciate it if you considered recommending our paper to be presented at the conference.

---

> > > > ### Author Response · Authors · 2025-11-26
> > > > **Follow-up discussion**
> > > >
> > > > Dear Reviewer 8aY8,
> > > >
> > > > As the discussion period is coming to a close, we would like to kindly follow up. We have taken great care to address your questions and concerns, and if you have any further questions or comments regarding our work, we would be more than happy to discuss and respond before the deadline.
> > > >
> > > > Thank you again for your time and valuable feedback!

---

### Official Review · Reviewer_DdHb · 2025-11-05

**Soundness:** 3
**Presentation:** 2
**Contribution:** 3
**Rating:** 4
**Confidence:** 3

**Summary:**

This paper addresses the high computational cost of training quantization modules for state-of-the-art VQ-based visual tokenizers by proposing VQ-Transplant, a framework enabling plug-and-play replacement of VQ modules in pre-trained tokenizers without costly end-to-end retraining. it also introduces MMD-VQ, a novel VQ algorithm using Maximum Mean Discrepancy to align feature and codebook distributions, which eliminate Gaussian data assumption, for better compatibility with VQ-Transplant. VQ-Transplant achieves near state-of-the-art reconstruction fidelity on models like VAR, reduces training cost, and shows strong cross-dataset generalization on FFHQ, CelebA-HQ, and LSUN-Churches.

**Strengths:**

1. VQ-Transplant gives efficiently an admirable performance with various VQ method. In Table 1 demonstrate a comparison over training cost among different work, which shows a large amount of time saving.
2. MMD-VQ eliminate gaussian data assumption, which shows a certain of potential.
3. MMD-VQ shows a competable performance as Wasserstein VQ and shows an ability of generalization to other domain.

**Weaknesses:**

1. VQ-Transplant is only experimentally valid on VAR. The framework is exclusively validated on the VAR pre-trained tokenizer, with no tests on other mainstream visual tokenizers (e.g., VQGAN, LDM) to verify its compatibility across architectures.
2. While the VQ transplant framework with frozen encoder weights is designed to enable the plug-and-play integration of new VQ modules into frozen, pre-trained tokenizers, it still lacks a full-scale training experiment. Without such an experiment, a direct performance-versus-time comparison between full-scale training and VQ transplant cannot be established. Consequently, it remains unclear whether the proposed framework delivers a superior performance-time ratio.

**Questions:**

1. In most experiments, Wasserstein VQ and MMD VQ exhibit quite comparable performance. Given this, what are the practical advantages of MMD VQ over the former? Or in another way, can you prove that MMD-VQ does provide a better performance on non-gaussian data than Wasserstein VQ?
2.  Since VQ-Transplant and MMD VQ are two key contributions of this paper, consideration should be given to dividing these two components into separate subsections in Chapter 4. Although you claimed MMD-VQ is a method introduced within VQ-Transplant, MMD-VQ can be implemented without VQ-Transplant. Describing these two contributions in a more separate way can help understand the subsequent experimental results.
3. In line 340 there seems to be a mistake. "Table 3 tracks the progression of r-FID metrics throughout decoder adaptation on ImageNet-1K across training epochs.", "table 3" should be "table 4".

---

> ### Author Response · Authors · 2025-11-23
> **Response to Reviewer DdHb (1/4)**
>
> Thank you for taking the time to review our paper and for your thoughtful feedback. We address your questions and concerns below.
>
> ---
>
> > R 1.1 VQ-Transplant is only experimentally valid on VAR. The framework is exclusively validated on the VAR pre-trained tokenizer, with no tests on other mainstream visual tokenizers (e.g., VQGAN, LDM) to verify its compatibility across architectures.
>
> Thank you for raising this important point regarding the applicability of VQ-Transplant beyond the VAR tokenizer.
>
> ### **(1) Compatibility with Other Pretrained Tokenizers**
>
> To evaluate generality, we applied VQ-Transplant to the pretrained LDM-16 continuous tokenizer (see Appendix D). As Table 16 (or below) shows, VQ-Transplant achieves reasonable performance on LDM-16. Nevertheless, its adaptability is lower compared to VAR-based models, as compared in Table 3, particularly in terms of r-FID and r-IS metrics.
>
> | Methods | Tokens | Codebook Size | $\mathcal{E}$ | $\mathcal{U}$ | PSNR | SSIM | LPIPS | r-FID | r-IS |
> | --- | --- | --- | --- | --- | --- | --- | --- | --- | --- |
> | LDM-16 | 256 | - | 0.0 | - | 24.08  | **68.0** | - | **0.87** | **210.3** |
> | Vanilla VAR | 680 | 4096 | 0.424  | 97.0\% | 22.66 | 56.0 | 0.157 | 4.89 | 136.7 |
> | EMA VAR | 680 | 4096 | 0.367 | **100\%** | 22.88 | 57.0 | 0.147 | 4.11 | 144.9 |
> | Online VAR | 680 | 4096 | 0.299 | **100\%** | 23.45 | 59.1 | 0.134 | 3.18 | 158.8 |
> | Wasserstein VAR | 680 | 4096 | **0.278** | **100\%** | 23.49 | **59.5** | **0.130** | **2.87** | **161.4** |
> | MMD VAR | 680 | 4096 | **0.278** | **100\%** | **23.51** | **59.5** | 0.131 | 2.93 | 161.2 |
> | Vanilla VAR | 680 | 8192 | 0.418 | 87.3\% | 22.58 | 56.0 | 0.157 | 5.14 | 132.6 |
> | EMA VAR | 680 | 8192 | 0.333 | **100\%** | 23.26 | 58.4 | 0.139 | 3.59 | 151.6 |
> | Online VAR | 680 | 8192 | 0.283 | 94.7\% | 23.57 | 59.6 | 0.130 | 2.98 | 159.9 |
> | Wasserstein VAR | 680 | 8192 | **0.252** | **100\%** | 23.59 | 60.0 | **0.125** | **2.58** | **166.9** |
> | MMD VAR | 680 | 8192 | 0.254 | **100\%** | **23.68** | **60.4** | **0.125** | 2.68 | 166.2 |
>
> This performance gap is likely attributable to two factors
>
> - **Model capacity difference:** The VAR tokenizer has a larger encoder–decoder (104M parameters) compared to LDM-16 (68M), providing stronger reconstruction
> capabilities and greater flexibility during adaptation.
> - **Decoder adaptation behavior:** The VAR decoder is pretrained to handle quantized features $z_q$, which naturally makes it robust to the deviations introduced by new VQ algorithms. In contrast, the LDM decoder only sees continuous features $z_e$ during training, making it less adaptable to different VQ modules. This explains why VAR-based VQ-Transplant achieves strong results with few adaptation epochs, while LDM may require more training.
>
> ### **(2) Key Contributions Beyond VAR**
>
> Even though VQ-Transplant relies on **a well-trained discrete tokenizer like VAR, our contributions remain significant**:
>
> - **Broader usability of VAR:** The VAR repository has not yet released the full tokenizer training code (check VAR GitHub repository issue #125, #123, #28, #175, etc) and only provides fixed checkpoints (codebook size 4096) trained on OpenImages. VQ-Transplant enables researchers to adapt the VAR tokenizer to different codebook sizes (e.g., 8192 or larger) and datasets (e.g., FFHQ, CelebA, Church), thereby democratizing access to high-quality discrete tokenization.
> - **Lowering the computational barrier:** VQ-Transplant enables competitive performance with minimal resources, making discrete tokenization research more accessible, e.g., to academia and public institutions.
> - **Improved performance with advanced VQ algorithms:** By combining with advanced VQ algorithms, as shown in below table, VQ-Transplant can surpass the original VAR performance (e.g., rFID improvements with MMD-VAR after adaptation epochs).
>
> | Methods | Tokens | Codebook Size | $\mathcal{E}$ | r-FID | r-IS |
> | --- | --- | --- | --- | --- | --- |
> | VAR | 680 | 4096 | 0.283 | 0.92 | 198.6 |
> | MMD VAR-a  | 680 | 4096 | 0.255 | 0.91 | 199.2 |
> | MMD VAR-a | 680 | 8192 | 0.234 | 0.81 | **201.0** |
> | MMD VAR-b | 680 | 4096 | 0.255 | 0.87 | **199.9** |
> | MMD VAR-b | 680 | 8192 | 0.234 | 0.78 | **201.0** |
> | MMD VAR-c | 680 | 4096 | 0.255 | 0.82 | 198.3 |
> | MMD VAR-c | 680 | 8192 | 0.234 | 0.75  | **201.0** |
> | MMD VAR-d | 680 | 4096 | 0.255 | **0.79**  | 199.1 |
> | MMD VAR-d | 680 | 8192 | 0.234 | **0.74**  | **201.0** |
>
> The suffixes “–a”, “–b”, “–c”, and “–d” correspond to decoder adaptation for 5, 10, 15, and 20 epochs, respectively.
>
> **Summary**
>
> While VQ-Transplant’s current efficiency is most evident with VAR due to its pretrained robustness, it can in principle be applied to other tokenizers with sufficient adaptation. More importantly, our framework unlocks practical and performance improvements that were previously inaccessible, which addresses a strong community demand.
>
> ---

---

> > ### Author Response · Authors · 2025-11-23
> > **Response to Reviewer DdHb (2/4)**
> >
> > > R 1.2 While the VQ transplant framework with frozen encoder weights is designed to enable the plug-and-play integration of new VQ modules into frozen, pre-trained tokenizers, it still lacks a full-scale training experiment. Without such an experiment, a direct performance-versus-time comparison between full-scale training and VQ transplant cannot be established. Consequently, it remains unclear whether the proposed framework delivers a superior performance-time ratio.
> >
> > Thank you for raising this point. To address this, we conducted a set of full-scale training experiments. In the table below, the suffixes “–a,” “–b,” and “–c” correspond to full-scale training for 5, 6, and 7 epochs, respectively (more results are provided in Table 6). These results empirically confirm that VQ-Transplant achieves superior performance-to-training-time efficiency.
> >
> > | Methods | Training Strategy | GPUs | Training Hours | Codebook Size | r-FID | r-IS |
> > | --- | --- | --- | --- | --- | --- | --- |
> > | MMD-VAR | VQ-Transplant | 2 $\times$ A100 | 22 h | 4096 | **0.91** | **199.2** |
> > | MMD VAR  | VQ-Transplant | 2 $\times$ A100 | 22 h | 8192 | **0.81** | **201.0** |
> > | MMD VAR-a | Full-scale Training | 2 $\times$ A100 | 25 h | 4096 | 1.40 | 189.4  |
> > | MMD VAR-a | Full-scale Training | 2 $\times$ A100 | 25 h | 8192 | 1.26  | 191.0 |
> > | MMD VAR-b | Full-scale Training | 2 $\times$ A100 | 30 h | 4096 | 1.36 | 191.9 |
> > | MMD VAR-b | Full-scale Training | 2 $\times$ A100 | 30 h | 8192 | 1.29  | 190.2 |
> > | MMD VAR-c | Full-scale Training | 2 $\times$ A100 | 35 h | 4096 | 1.34 | 190.4 |
> > | MMD VAR-c | Full-scale Training | 2 $\times$ A100 | 35 h | 8192 | 1.26  | 191.0 |
> >
> > Even when full-scale training runs longer than VQ-Transplant, its reconstruction performance remains substantially lower. This outcome is expected because discrete tokenizers typically require hundreds of epochs to achieve high-quality visual reconstruction **when trained from scratch**. In contrast, VQ-Transplant leverages **pretrained encoder features**, allowing it to reach strong reconstruction quality in far fewer epochs.
> >
> > Overall, VQ-Transplant **not only reduces training hours and computational cost**, but also delivers better performance in a fraction of the time, clearly demonstrating a superior performance-to-time trade-off compared with full-scale training.

---

> ### Author Response · Authors · 2025-11-23
> **Response to Reviewer DdHb (3/4)**
>
> > R 1.3 In most experiments, Wasserstein VQ and MMD VQ exhibit quite comparable performance. Given this, what are the practical advantages of MMD VQ over the former? Or in another way, can you prove that MMD-VQ does provide a better performance on non-gaussian data than Wasserstein VQ?
>
> This is an insightful question. We address it from both theoretical and empirical perspectives.
>
> ### **(1) Theoretical Motivation of MMD-VQ**
>
> Maximum Mean Discrepancy (MMD) is motivated by a fundamental principle in vector quantization (VQ): **VQ is near-optimal when the codebook distribution matches the underlying feature distribution, minimizing quantization error and maximizing codebook utilization**. Prior works [1][2] provide rigorous support for this principle. Building on this, [1] propose using Wasserstein distance (WD) to align distributions.
>
> The key distinction between MMD-VQ and Wasserstein VQ lies in assumptions about the feature distribution:
>
> - **Wasserstein-VQ**: For computational tractability, WD often relies on a Gaussian assumption. Specifically, for $p=2$, the 2-Wasserstein distance between two Gaussians $\mathcal{N}(\mu_1, \Sigma_1)$ and $\mathcal{N}(\mu_2, \Sigma_2)$ admits a closed-form:
>
>     \begin{equation} W_{2}(\mathcal{N}(\mu_1, \Sigma_1), \mathcal{N}(\mu_2, \Sigma_2)) = \sqrt{\Vert \mu_1 - \mu_2 \Vert^2_2 + \mathop{\mathrm{tr}}\left( {\Sigma_1}+ {\Sigma_2} - 2 (\Sigma_1^{\frac{1}{2}} {\Sigma_2} \Sigma_1^{\frac{1}{2}})^{\frac{1}{2}}\right)}. \end{equation}
>
>
>    This effectively aligns only the **first- and second-order moments** of the distributions. For approximately Gaussian features, this is often sufficient. However, for non-Gaussian or multi-modal features, such moment matching is inherently limited.
>
> - **MMD-VQ**: MMD leverages RKHS embeddings with characteristic kernels (e.g., Gaussian RBF), which theoretically match **all moments** of the distributions. This provides a more expressive and flexible criterion for distribution matching, enabling better alignment for non-Gaussian features. More comprehensive discussion are provided in Appendix B.
>
> **References:**
>
> [1] Enhancing Vector Quantization with Distributional Matching: A Theoretical and Empirical Study
>
> [2] Foundations of Quantization for Probability Distributions
>
> ### **(2) Why MMD-VQ and Wasserstein-VQ Show Comparable Empirical Performance**
>
> In standard benchmarks (ImageNet, FFHQ, CelebA-HQ, LSUN-Churches), encoder-produced latent features are typically near-Gaussian, likely due to aggregation of many weakly dependent components (a phenomenon consistent with the Central Limit Theorem). In such cases, aligning the first two moments suffices, and MMD-VQ’s advantages are less pronounced, explaining the comparable performance.
>
> ### **(3) Advantage of MMD-VQ on Non-Gaussian Data**
>
> To test cases where WD’s Gaussian assumption fails, we construct a **synthetic non-Gaussian latent distribution** as a bimodal mixture (More details and results see Appendix B):
>
> \begin{equation}  0.5 \cdot \mathcal{N}(\zeta \mathbf{1}, I) + 0.5 \cdot \mathcal{N}(-\zeta  \mathbf{1}, I) , \end{equation}
>
> where $\zeta \in {0.0, 0.5, \dots, 4.0}$ controls deviation from Gaussianity. Larger $\zeta$ corresponds to a higher level of non-Gaussianity.
>
> We evaluate Wasserstein-VQ and MMD-VQ in terms of **quantization error** and **codebook utilization**:
>
> Quantization error:
>
> | **Methods** | 0.0 | 0.5 | 1.0 | 1.5 | 2.0 | 2.5 | 3.0 | 3.5 | 4.0 |
> | --- | --- | --- | --- | --- | --- | --- | --- | --- | --- |
> | Wasserstein VQ | 0.976 | 1.099 | 1.177 | 1.252 | 1.318 | 1.373 | 1.420 | 1.462 | 1.502 |
> | MMD VQ | 0.968 | 1.088 | 1.142 | 1.155 | 1.171 | 1.186 | 1.198 | 1.217 | 1.240 |
>
> Codebook utilization:
>
> | Methods | 0.0 | 0.5 | 1.0 | 1.5 | 2.0 | 2.5 | 3.0 | 3.5 | 4.0 |
> | --- | --- | --- | --- | --- | --- | --- | --- | --- | --- |
> | Wasserstein VQ | 99.9\% | 99.8\% | 97.0\% | 78.2\% | 62.7\% | 52.1\% | 44.8\% | 39.2\% | 34.8\% |
> | MMD VQ | 99.9\% | 99.9\% | 99.8\% | 97.0\% | 92.5\% | 88.9\% | 85.7\% | 81.2\% | 75.6\% |
>
> Observations:
>
> - For small $\zeta$ (nearly Gaussian), Wasserstein-VQ and MMD-VQ perform comparably.
> - As $\zeta$ increases (more non-Gaussian), MMD-VQ consistently achieves lower quantization error and better codebook utilization, whereas Wasserstein-VQ degrades due to its limited moment-matching assumption.
>
> These synthetic experiments **empirically confirm that MMD-VQ provides a practical advantage when the feature distribution is non-Gaussian**, validating the theoretical expectation.
>
> **Summary:**
>
> - MMD-VQ is strictly more flexible and expressive for distribution matching.
> - On real-world benchmarks with near-Gaussian features, differences are small.
> - On non-Gaussian or multi-modal features, MMD-VQ clearly outperforms Wasserstein-VQ in both quantization error and codebook utilization.

---

> ### Author Response · Authors · 2025-11-23
> **Response to Reviewer DdHb (4/4)**
>
> **Supplementary Results: Advantage of MMD-VQ on Non-Gaussian Data (for R1.3)**
>
> For the non-Gaussian setting with $\zeta = 4.0$, we extend the training horizon from 10k to 100k steps to ensure both methods fully converge. Under this longer training schedule, both approaches reach stable solutions, and the performance gap becomes even more pronounced in favor of MMD-VQ, consistently improving on both quantization error and codebook utilization.
>
> Quantization error ($\zeta=4.0$):
> | Training Steps | 10k | 20k | 30k | 40k | 50k | 60k | 70k | 80k | 90k | 100k |
> | --- | --- | --- | --- | --- | --- | --- | --- | --- | --- | --- |
> | Wasserstein VQ | 1.502 | 1.462 | 1.447 | 1.432 | 1.418 | 1.412 | 1.403 | 1.398 | 1.396 | 1.394 |
> | MMD VQ | 1.240 | 1.188 | 1.177 | 1.169 | 1.161 | 1.162 | 1.157 | 1.156 | 1.156 | 1.157 |
>
> Codebook utilization ($\zeta=4.0$):
>
> | Training Steps | 10k | 20k | 30k | 40k | 50k | 60k | 70k | 80k | 90k | 100k |
> | --- | --- | --- | --- | --- | --- | --- | --- | --- | --- | --- |
> | Wasserstein VQ | 34.8\% | 37.2% | 38.8% | 39.9% | 40.6% | 41.4% | 41.9% | 42.5% | 43.0% | 43.4% |
> | MMD VQ | 75.6\% | 85.9% | 89.5% | 91.4% | 92.4% | 92.9% | 93.3% | 93.6% | 93.9% | 94.0% |
>
> These additional results reinforce the main claim: MMD-VQ not only converges reliably on non-Gaussian data but also achieves substantially lower quantization error and significantly higher codebook utilization, even under long-horizon training.
>
> ---
>
> > R 1.4 Since VQ-Transplant and MMD VQ are two key contributions of this paper, consideration should be given to dividing these two components into separate subsections in Chapter 4. Although you claimed MMD-VQ is a method introduced within VQ-Transplant, MMD-VQ can be implemented without VQ-Transplant. Describing these two contributions in a more separate way can help understand the subsequent experimental results.
>
> We appreciate the comment regarding the presentation of VQ-Transplant and MMD-VQ in Chapter 4. We fully agree that separating these two contributions into distinct subsections will improve clarity and help readers better understand their respective roles and effects.
>
> Although MMD-VQ was introduced within the VQ-Transplant framework, it is indeed a standalone method that can be implemented independently, and it is not the core contribution of our submission. To avoid confusion and clearly attribute experimental results, we **will revise Chapter 4 to include separate, dedicated subsections** for VQ-Transplant and MMD-VQ. Specifically:
>
> - The **VQ-Transplant** subsection will focus on the method’s training and transfer strategy, highlighting its practical benefits and experimental impact. This is the main contribution of this work.
> - The **MMD-VQ** subsection will detail the distribution alignment approach, its motivation, and how it contributes to improved quantization performance.
>
> We believe this reorganization will make the paper more readable, allow readers to clearly distinguish the contributions of each component, and ensure that the effects of VQ-Transplant and MMD-VQ on downstream experiments are fully appreciated.
>
> ---
>
>
> Please let us know if you have any remaining questions. If our clarifications and additional experiments have satisfactorily addressed your questions and concerns, we would deeply appreciate it if you considered recommending our paper to be presented at the conference.

---

> > ### Author Response · Authors · 2025-11-26
> > **Follow-up discussion**
> >
> > Dear Reviewer DdHb,
> >
> > As the discussion period is coming to a close, we would like to kindly follow up. We have taken great care to address your questions and concerns, and if you have any further questions or comments regarding our work, we would be more than happy to discuss and respond before the deadline.
> >
> > Thank you again for your time and valuable feedback!

---

### Author Response · Authors · 2025-11-23
**General Responses to All Reviewers**

Thank you to all the reviewers for their valuable feedback. We sincerely appreciate your constructive comments. We are grateful to **reviewers DdHb and auQc** for highlighting the practical significance and efficiency of VQ-Transplant, as well as its strong cross-dataset generalization. We also thank **reviewers 8aY8 and tdwt** for recognizing the plug-and-play flexibility of our framework and the novelty of MMD-VQ in aligning feature and codebook distributions without Gaussian assumptions. Finally, we appreciate all reviewers for noting the solid empirical validation and overall clarity of our work.

In response to the reviewers’ feedback, we have made updates and several additions to the manuscript, summarized as follows:

- **Clear Motivation for MMD-VQ:** In Appendix B, we provide a detailed theoretical motivation for MMD-VQ, extending the distribution alignment framework of Wasserstein-based vector quantization to Maximum Mean Discrepancy. We also empirically demonstrate the advantages of MMD-VQ over Wasserstein VQ on non-Gaussian data, as shown in Tables 12 and 13 and Figure 7.
- **Compatibility of VQ-Transplant with Pretrained LDM Tokenizer:** In Appendix D, we examine the compatibility of VQ-Transplant with the continuous LDM tokenizer, with experimental results presented in Table 16.
- **Joint Optimization of Encoder, Decoder, and VQ in Stage II:** In Appendix C, we discuss an alternative optimization strategy in which the encoder, decoder, and VQ module are jointly optimized. The corresponding experimental results are shown in Table 14.
- **Comparison Between From-scratch Training and VQ-Transplant:** We empirically compare training from scratch with VQ-Transplant under roughly comparable training costs, as reported in Table 6.
- **Extended Decoder Adaptation Epochs:** We further extended the decoder adaptation to 20 epochs. The results are reported in Table 5, and Figure 3 presents the corresponding r-FID curves.

Once again, we sincerely thank the reviewers for their thoughtful comments. We have carefully addressed each point in our revised manuscript to clarify the motivation, effectiveness, and generality of our proposed framework.

---

### Author Response · Authors · 2025-12-04
**Message to the Area Chair - Summary of Discussion**

Dear Area Chair,

In this overview, we summarize the key comments by the reviewers and our responses to them. Thank you for your time and consideration.

---

We appreciated that **Reviewer DdHb** recognized the efficient and strong performance of VQ-Transplant across various VQ methods with significant training cost savings, as well as the potential of MMD-VQ in eliminating Gaussian assumptions, achieving competitive performance comparable to Wasserstein VQ, and demonstrating the ability to generalize across domains.

We were delighted that **Reviewer 8aY8** recognized the general idea of enabling plug-and-play components to explore a space of algorithmic inductive biases, as well as the interesting and potentially novel proposal of MMD-VQ.

We were pleased that **Reviewer auQc** recognized the novel and practical idea of transplanting VQ modules into frozen tokenizers to enable flexible experimentation, the principled formulation of MMD-VQ for aligning feature and codebook distributions without Gaussian assumptions, and the strong empirical performance demonstrating both effectiveness and substantial reductions in training cost across multiple datasets and architectures.

We were happy that **Reviewer tdwt** recognized that our method is well written, plug-and-play, applicable to any pretrained discrete image tokenizer, and effectively reduces the training time of the codebook.

---

In our responses to individual reviewers, we have carefully addressed all questions and concerns in great detail, providing clarification where needed and presenting additional, requested empirical results. We provide a short summary of our responses to the reviewers below.

---

> ### Author Response · Authors · 2025-12-04
> **Discussion with Reviewer tdwt**
>
> **Discussion**
>
> **Reviewer tdwt** originally shared five concerns, engaged in discussion with us during the early discussion phase, and **stated that all of their concerns had been satisfactorily addressed**. **The score was raised to 6.**
>
> ---
>
> The reviewer’s five concerns include: the request for an extended comparison table (**R 4.1**); the observation that MMD-VAR’s reported performance appears comparable to VAR (**R 4.2**); the concern that VQ-Transplant may simply reproduce the original VAR codebook (**R 4.3**); the lack of evaluation of generative capabilities (**R 4.4**); and the recommendation to test on continuous VAEs (**R 4.5**).
>
> Among these, **R 4.2** and **R 4.3** are the most significant concerns, and we addressed them from both a theoretical and an empirical angle.
>
> - **Theoretical Perspective:** MMD-VAR builds on established vector quantization principles: A VQ model is near-optimal when the codebook distribution aligns with the feature distribution, minimizing quantization error and maximizing codebook utilization. MMD-VAR enforces this alignment by minimizing the MMD distance with characteristic kernels, ensuring universal distribution matching. Although the original VAR codebook training is not publicly disclosed, MMD-VAR’s principled approach achieves near-optimal adaptation, yielding improved performance. The learned codebook is thus a refined improvement, not a simple replication.
>
> - **Empirical Perspective:** Table 3 shows that after two adaptation epochs of VQ module substitution, MMD-VAR attains lower quantization error than VAR (0.255 vs. 0.283), indicating reduced information loss. Thus, at the VQ module level, MMD-VAR outperforms VAR. Stage I reconstruction is initially lower (rFID: 1.52 vs. 0.92) because the VAR decoder was trained for larger quantization errors, while MMD-VAR introduces smaller errors. After sufficient decoder adaptation epochs, the advantage of lower quantization error ultimately translates into improved reconstruction performance (0.79), surpassing VAR (0.92) at the same codebook size.
>
> In summary, both theoretically and empirically, MMD-VAR provides a principled and effective enhancement of the VAR VQ module, with clear performance benefits emerging after decoder fine-tuning. The learned codebook represents a meaningful improvement rather than a mere replication of the original.
>
> Beyond addressing the reviewer’s main concerns, we also implemented the additional suggestions: we expanded the comparison table in **R 4.1** and conducted VQ-Transplant experiments on continuous VAE models (e.g., LDM-16) in **R 4.5**. We further explained that including full generative-model results is infeasible within the rebuttal window due to the community-recognized **extremely high training cost** and the **limited reproducibility** of current VAR-style generative pipelines in **R 4.4**.
>
> ---
>
> **Conclusion**
>
> Thanks to these comprehensive rebuttal efforts, **Reviewer tdwt** concluded that **all of his concerns have been fully resolved** and **raised their score to 6**.

---

> ### Author Response · Authors · 2025-12-04
> **Discussion with Reviewer 8aY8**
>
> **Discussion**
>
> Comments **R 2.1** and **R 2.2** below appeared to be the reviewer’s primary focus. We provided extensive evidence to highlight our contributions and clarify any misunderstandings in our response. We summarize the main points of our response below.
>
> ---
>
> > **R 2.1**: VQ-Transplant may resemble the standard practice of using a pretrained vision model as a frozen feature extractor and training a new classification layer, which could reintroduce representational biases; the reviewer suggested fine-tuning the encoder.
>
> We emphasized our contributions in three key aspects:
>
> - **Democratizing access to VAR**: The VAR repository has not yet released the full tokenizer training code; only fixed checkpoints (codebook size 4096, trained on OpenImages) are available, making retraining on custom datasets or alternative codebook sizes infeasible. VQ-Transplant enables adaptation to datasets such as FFHQ, CelebA, and LSUN Church, and allows scaling the codebook (e.g., to 8192) without retraining from scratch.
>
> - **Reducing computational barriers**: Training a modern tokenizer like VAR typically requires adversarial training and long schedules, making replication or modification costly. VQ-Transplant achieves comparable or superior performance at minimal computational cost (<5% of the original), significantly lowering the barrier to access.
>
> - **Enhanced performance via advanced VQ algorithms**: Incorporating improved VQ losses (e.g., MMD-VQ) allows the transplanted tokenizer to surpass the reconstruction performance of the original VAR.
>
> In short, VQ-Transplant goes beyond the conventional “frozen encoder + new head” paradigm, efficiently satisfying the community’s needs with minimal computational cost.
>
> Second, we additionally compared VQ-Transplant with training a tokenizer from scratch and found that **under comparable or even lower training costs, VQ-Transplant consistently achieves superior reconstruction quality**. While initializing from a pretrained tokenizer may introduce some representational bias, our experiments demonstrate that VQ-Transplant **allows advanced VQ algorithms to achieve state-of-the-art results efficiently**.
>
> Third, following the reviewer’s suggestion, we also conducted experiments with encoder fine-tuning. Joint optimization further improves performance, with an additional 34% training cost.These results fully address all aspects of **R 2.1**.
>
> ---
>
> > **R 2.2**: The paper requires a more detailed analysis of MMD-VQ, with a subjective concern regarding potential mode collapse.
>
> > **R 2.3**: Requests clarification on MMD-VQ implementation details.
>
> We provided both theoretical and empirical evidence for MMD-VQ:
>
> - **Theoretical motivation and empirical validation**: As detailed in Appendix B, MMD-VQ is theoretically motivated and demonstrates clear advantages on non-Gaussian data.
>
> - **Encoder fine-tuning**: Jointly optimizing the encoder with MMD-VQ **does not induce mode collapse**; instead, it improves reconstruction performance, directly addressing the reviewer’s concern.
>
> - **Implementation details**: To avoid potential mode collapse, we **detach latent features when computing the MMD loss**, preventing the encoder distribution from being pulled toward the codebook distribution.
>
> Taken together, these results provide strong evidence that MMD-VQ is both theoretically sound and empirically robust.
>
> ---
>
> **Conclusion**
>
> We carefully and exhaustively addressed each of **Reviewer 8aY8**’s concerns, providing both **theoretical justification and empirical validation**.

---

> ### Author Response · Authors · 2025-12-04
> **Discussion with Reviewer DdHb**
>
> **Discussion**
>
> > **R 1.1**: VQ-Transplant has only been experimentally validated on VAR, with no evaluation on other mainstream visual tokenizers (e.g., VQGAN, LDM) to verify compatibility across architectures.
>
> As suggested, we conducted additional experiments by applying VQ-Transplant to the pretrained LDM-16 tokenizer to verify its compatibility across architectures. This concern is **identical to R 4.5** raised by **Reviewer tdwt**, who acknowledged that it has been fully addressed. Therefore, we have strong reason to believe that **Reviewer DdHb’s concern has also been fully resolved**.
>
> ---
>
> > **R 1.2**: The absence of full-scale training experiments prevents a direct performance–time comparison with VQ-Transplant.
>
> As suggested, we conducted full-scale training experiments. The results confirm that VQ-Transplant achieves **a clearly superior performance–time trade-off**: it requires **far fewer training hours and computational resources** while delivering **stronger performance**.
>
> ---
>
> > **R 1.3**: Whether MMD-VQ offers any practical or provable advantage over Wasserstein-VQ, particularly on non-Gaussian data, given their similar empirical performance.
>
> - VQ is near-optimal when the codebook distribution matches the underlying feature distribution, which minimizes quantization error and maximizes codebook utilization. Both Wasserstein VQ and MMD VQ aim to achieve such distribution matching.
> - Wasserstein VQ aligns only **the first- and second-order moments**, whereas MMD VQ theoretically matches **all moments** of the distributions.
> - On standard benchmarks, encoder-produced latent features are typically **close to Gaussian**, likely due to the Central Limit Theorem. In such cases, aligning the first two moments is sufficient, explaining the comparable performance observed.
> - We construct a synthetic **non-Gaussian latent distribution** using a bimodal mixture. These experiments empirically confirm that MMD VQ offers a practical advantage when the feature distribution is non-Gaussian, consistent with theoretical expectations.
>
> ---
>
> **Conclusion**
>
> We have carefully addressed each of **Reviewer DdHb**’s concerns, providing **both theoretical justification and empirical validation**.

---

> ### Author Response · Authors · 2025-12-04
> **Discussion with Reviewer auQc**
>
> **Discussion**
>
> We greatly appreciated **Reviewer auQc's strong endorsement of our contributions**, reflected in their **score of 8**, as well as their thoughtful comments.
>
> **Reviewer auQc** did not raise any major concerns but rather focused on questions regarding potential extensions. We address these comments below.
>
> ---
>
> > **R 3.1** Limited scope of tokenizer architectures, as experiments are restricted to VAR-based tokenizers.
>
> As suggested, we conducted additional experiments by applying VQ-Transplant to the pretrained LDM-16 tokenizer to verify its compatibility across architectures. This concern is **identical to R 4.5** raised by **Reviewer tdwt**, who acknowledged that it has been fully addressed. Therefore, we have strong reason to believe that **Reviewer auQc’s concern has also been fully resolved**.
>
> ---
>
> > **R 3.2 & 3.3** Limited evaluation in image generation and multimodality; questions regarding extending MMD-VQ to video or multimodal quantization.
>
> We clarified that we are unable to conduct these additional experiments within the rebuttal period and that extending our work to multimodal settings is beyond the scope of the current paper, but we intend to pursue this direction in future work. Importantly, MMD-VQ is **modality-agnostic** and can naturally be applied to video and multimodal tasks. To avoid encoder collapse when applying MMD-VQ to new domains, we suggest to **detach the latent features** during MMD loss computation, ensuring stable training across different modalities.
>
> ---
>
> **Conclusion**
>
> In summary, each of **Reviewer auQc**’s points has been fully addressed, either empirically (**R 3.1**) or conceptually with a clear description of potential future directions (**R 3.2 & R 3.3**).

---

### Meta-Review · Area_Chair_1UKd · 2025-12-05

**Summary:**

One key contribution of the paper is MMD VQ, but reviewer raised concern about Wasserstein VQ and MMD VQ regarding quite comparable performance exhibited by them. Authors attribute this to Gaussian distribution. To rebut, experiments on the synthetic experiments are performed, but  they still claim that MMD VQ provides a practical advantage, which is not very convincing based on the synthetic data. However, this work still provides a solid validation and their proposal, which is meaningful to the quantified codebook. Thus I recommend this paper.

**Reviewer Concerns:**

The reviewers’ main concerns focused on whether the proposed method can generalize to broader frameworks and how it differs from conventional pretrain–finetune pipelines. One reviewer notice two severe limitations. First, the authors indicated that VQ-transplant is a plug-and-play operation, but did not validate it on other VQ models. The framework's core value lies in its generalizability among VQ architectures. Without verification on more VQ models, its generalizability remains unproven. Second, the manuscript lacks full fine-tuning experiments. This is critical because the authors claim that VQ-Transplant is computationally efficient. Without full fine-tuning experiments, the claim is unbacked. These two limitations are related to the authors' fundamental claims (generalizability and performance-time balance).

**Reviewer Scores:**

Reviewer tdWt has already indicated an increased score of 6. Reviewer 8aY8 keep the initial score of 2 makes it uncertain whether remaining concerns might persist, but the reviewer agreed with the contribution of MMD. Reviewer auQc started with a score of 8 but withe lower confidence. Reviewer DdHb received detailed responses to all concerns, but further raise more concerns about the key contributions and keep the score 4.

---

### Decision · Program_Chairs · 2026-01-26

Accept (Poster)